# Framework for developing hybrid process-driven, artificial neural network and regression models for salinity prediction in river systems

Jason M. Hunter[1], Holger R. Maier[1], Matthew S. Gibbs[1,2], Eloise R. Foale[1], Naomi A. Grosvenor[1], Nathan P. Harders[1], Tahali C. Kikuchi-Miller[1],

[1]School of Civil, Environmental and Mining Engineering, The University of Adelaide, Adelaide, 5005 SA, Australia
[2]Department of Environment, Water and Natural Resources, GPO Box 2384, Adelaide, 5001 SA, Australia

*Correspondence to*: Jason M. Hunter (jason.hunter@adelaide.edu.au)

**Abstract.** Salinity modelling in river systems is complicated by a number of processes, including in-stream salt transport and various mechanisms of saline accession that vary dynamically as a function of water level and flow, often at different temporal scales. Traditionally, salinity models in rivers have either been process- or data-driven. The primary problem with process-based models is that in many instances, not all of the underlying processes are fully understood or able to be represented mathematically, and that there are often insufficient historical data to support model development. The major limitation of data-driven models, such as artificial neural networks (ANNs), is that they provide limited system understanding and are generally not able to be used to inform management decisions targeting specific processes, as different processes are generally modelled implicitly. In order to overcome these limitations, a generic framework for developing hybrid process and data-driven models of salinity in river systems is introduced and applied in this paper. As part of the approach, the most suitable sub-models are developed for each sub-process affecting salinity at the location of interest based on consideration of model purpose, degree of process understanding and data availability, which are then combined to form the hybrid model. The approach is applied to a 46 km reach of the River Murray in South Australia, which is affected by high levels of salinity. In this reach, the major processes affecting salinity include in-stream salt transport, accession of saline groundwater along the length of the reach and the flushing of three waterbodies in the floodplain during overbank flows of various magnitudes. Based on trade-offs between the degree of process understanding and data availability, a process-driven model is developed for in-stream salt transport, an ANN model is used to model saline groundwater accession and three linear regression models are used to account for the flushing of the different floodplain storages. The resulting hybrid model performs very well on approximately three years of daily validation data, with a Nash-Sutcliffe efficiency (NSE) of 0.89 and a root mean squared error (RMSE) of 12.62 mg L$^{-1}$ (over a range from approximately 50 to 250 mg L$^{-1}$). Each component of the hybrid model results in noticeable improvements in model performance corresponding to the range of flows for which they are developed. The predictive performance of the hybrid model is significantly better than that of a benchmark process-driven model (NSE = -0.14, RMSE = 41.10 mg L$^{-1}$, G$_{bench}$ index = 0.90) and slightly better than that of a benchmark data-driven (ANN) model (NSE = 0.83, RMSE = 15.93 mg L$^{-1}$, G$_{bench}$ index = 0.36). Apart from improved predictive performance, the hybrid model also has advantages over the ANN benchmark model in terms of increased capacity for improving system understanding and greater ability to support management decisions.

## 1 Introduction

Models are being used increasingly for the management of hydrological systems such as streamflow (e.g. Hsu et al., 2002; Shamseldin et al., 2002; Dessie et al., 2014; Yaseen et al., 2016; Gibbs et al., 2018), reservoir inflow (e.g. Tsai et al., 2014; Gragne et al., 2015; Chang and Tsai, 2016), floods (e.g. Quiroga et al., 2013; Alvarez-Garreton et al., 2015; Kasiviswanathan et al., 2016), baseflow (e.g. Corzo and Solomatine, 2007; Li et al., 2014), water level (e.g. Chang and Chang, 2006; Shiri et al., 2016), groundwater level (Adamowski and Chan, 2011; Mohanty et al., 2015; Chang et al., 2016b), evaporation (e.g.

Parasuraman et al., 2007; Guo et al., 2016; Kisi and Demir, 2016), stream temperature (e.g. Gallice et al., 2015), ecosystem services and response (e.g. Chang et al., 2013; Duku et al., 2015), raw-water quality (Zhang and Stanley, 1997) and a range of other water quality parameters (e.g. Pulido-Velazquez et al., 2015; Kisi and Parmar, 2016), such as suspended sediment (e.g. Mount et al., 2012; Duan et al., 2015), phosphate (e.g. Chang et al., 2016) and salinity (e.g. Maier and Dandy, 1966; Bowden et al., 2005b). Such models can take different forms, ranging from fully process-driven (Habib et al., 2007; Liu et al., 2007),

to conceptual (Clark et al., 2011; Fenicia et al., 2011; Kavetski and Fenicia, 2011), to data-driven (Maier and Dandy, 1996; Bowden et al., 2005a). Process-driven models are developed from the known physical process(es) in a system, which are represented mathematically. Conceptual models represent the key elements of a system and the hypothesised relationships between them, and data-driven models are developed purely on available data with limited or no knowledge of the physical process represented in the model structure (Maier et al., 2010). As pointed out by Mount et al. (2016), all of these model types

are part of a spectrum based on the degree of influence of derived hypothetic knowledge or empirical data in their development. Hypothetic influence can be high if the underlying processes are well understood, as processes can be both described mathematically and incorporated into system models. In contrast, poorly-understood processes may not be able to be described explicitly in mathematical form, so the degree of hypothetic influence is necessarily lower, as information contained within measured data of the underlying system behaviour has to be relied upon to a greater degree for model development (e.g. for

determination of the functional form of the model, as well as model calibration).

Apart from the degree to which underlying process are understood and can be represented mathematically, the most appropriate modelling approach is also a function of the extent to which available data can adequately support model development (Grayson and Blöschl, 2000). For example, the structure of models that are based on well-known underlying processes can be

highly complex with a large number of parameters to accommodate that understanding, and therefore require significant volumes of data, which may not be available, for calibration. However, it should be noted that system understanding and/or analytical techniques designed to identify dominant processes can be useful in reducing model complexity and the amount of data needed for calibration (e.g. Gibbs et al., 2012; Markstrom et al., 2016). Conversely, and somewhat counterintuitively, data-driven models might be more suitable in such situations, as they are generally designed to make best use of existing data.

In other words, in more hypothetically influenced models, data requirements are dictated by model structure, which is generally derived based on system understanding. In contrast, in data-driven models, model structure is a function of available data. This means that best use can be made of existing data without the need to collect additional data to meet the requirements of a pre-determined model structure, as is generally the case with process-driven models (Mount et al., 2016). However, this has the disadvantage that not all dominant processes might be represented in the resulting model.


The final factor that can affect the suitability of different model types for a particular application is the intended purpose of the model. For example, if the primary purpose of the model is to improve system understanding, a model with a higher degree of hypothetic influence is likely to be of more value, although data-driven models have also been shown to be able to provide some insight into underlying processes (e.g. Jain et al., 2004; Kingtson et al., 2006; Jain and Kumar, 2009; Mount et al., 2013).

If the primary purpose of the model is the evaluation of various management options, care needs to be taken that these options are represented as inputs to the model and that the impacts of the management options are represented as model outputs, which can be achieved using a variety of model types. In other words, input-output relationships of the relevant processes need to be included in the model, but this can be achieved with or without explicit modelling of the underlying physical processes. Finally, if the primary purpose of the model is forecasting, explicit representation of the underlying processes is generally less important

when compared with model accuracy, although if a purely data-driven model is used, care needs to be taken to ensure that the model is updated when faced with input patterns that lie outside those used for initial model calibration (see Bowden et al., 2012; Zheng et al., 2018).

When dealing with complex integrated hydrological systems, overall system behaviour is likely to be affected by a number of

different sub-processes. While some of these might be well understood and supported by sufficient data to enable them to be modelled explicitly, others might not. Consequently, a model that falls on a single point of the hypothetically-driven - data-driven hydrological modelling spectrum might not be best suited to addressing the problem under consideration. In order to allow the relative strengths of different types of models on the hydrological modelling spectrum to be utilised fully, the use of hybrid models has been suggested (Corzo and Solomatine, 2007; Corzo et al., 2009; Robertson & Sharp, 2013; Mount et al.,

2016; Humphrey et al., 2016). Such models combine modelling approaches that fall on different points of the modelling spectrum to enable the most appropriate degree of hypothetic and data influence to be utilised in the modelling of each sub-process.

While a number of studies have illustrated the potential benefit of hybrid models, they have generally been confined to rainfall-

runoff / streamflow modelling (e.g. Hsu et al., 2002; Wang et al., 2006; Noori and Kalin 2016; Zhang et al., 2016). In addition, these studies have focused on a particular hybrid model structure, rather than a generic framework that can be used to develop the most suitable hybrid model in different settings. In order to overcome the above shortcomings, the objectives of this paper are:

1. To introduce a generic, high level conceptual framework for the development of hybrid models for modelling salinity in river systems that uses a combination of model purpose, knowledge of underlying system processes, and type and amount of available data to provide guidance for the selection of the suitable sub-models, thereby enabling the most appropriate balance between hypothetic and data influence to be struck in their development. While the proposed approach is specific to salinity modelling, the underlying principles presented are likely to be more widely applicable. The modelling of salinity in river systems is selected as the focus of the approach, as:

   a. High levels of salinity are a major concern in many river systems around the world (Rengasamy, 2006), due to their potentially detrimental impacts on the growth of agricultural crops, vegetation, bacteria and algae (Hart et al., 1991; Maier and Dandy, 1996) and adverse effects on water quality, as well as the stability of freshwater and neighbouring ecosystems. In addition, high salinity levels can have significant negative financial consequences stemming from the ongoing expense of treating drinking water, pumping at groundwater interception wells and from diminishing agricultural returns (Moxey, 2012).

   b. Salinity in river systems is generally affected by a number of complex processes (Williams, 2001; Goss, 2003), and the degree to which these processes are understood varies significantly (Maier and Dandy 1996; Woods, 2015). For example, there is generally a good understanding of the processes involved in the transport of salt with discharge, as salt is a conservative constituent. However, understanding of the complex processes associated with the accession of additional salt loads into the main river channel is often limited, as they are generally influenced by multiple interacting factors (e.g. land use, historical inundation regime, surface water-groundwater interactions, abstraction / recharge processes). In addition, the data needed to support the development of different types of models is highly variable.

   c. Current efforts directed towards the modelling of salinity in river systems has generally relied on either process-driven (Banerjee et al., 2011; Habib et al., 2007; Liu et al., 2007; Woods, 2015) or data-driven (Maier and Dandy, 1996; Huang and Foo, 2002; Suen and Lai, 2013; Bowden et al., 2005a; Bowden et al., 2002; Kingston et al., 2005; Rath et al., 2017) approaches. This has resulted in a number of limitations, such as difficulties in modelling the accession of salt via groundwater, wetlands and floodplains (e.g. groundwater regime shifts and flushing) explicitly (Harrington et al., 2006), which in turn makes it difficult to understand the relative importance of the different sources of saline accessions and to assess the potential utility of some of the management options mentioned earlier.

2. To illustrate the application and test the utility of the framework by applying it to a reach of the River Murray in South Australia, as this is an area where improved salinity modelling for management purposes would be of significant benefit (Beecham et al., 2003).

The remainder of this paper is organised as follows: the proposed framework providing guidance for developing hybrid salinity models for river systems is introduced in Section 2, followed by the application of the approach to a case study in a South Australian reach of the River Murray in Section 3. Details of the development of the relevant models are given in Section 4, while the corresponding results are presented and discussed in Section 5. A summary and conclusions are provided in Section 6.

**2 Proposed framework**

**2.1 Overview**

An outline of the proposed high level conceptual framework for supporting the development of hybrid models of salinity in complex river systems is given in Fig. 1. As can be seen, a hybrid model that consists of a number of sub-models representing the different processes affecting river salinity (e.g. in-channel salt transport, overland flow, flushing of anabranches and saline
groundwater accession) is suggested. The key feature of the proposed framework is that it uses a combination of model purpose, process understanding, and data availability and suitability to identify the most appropriate sub-models for the various processes under consideration. This includes the degree to which a particular process can be described mathematically and the suitability of the available data for the development of different types of models (e.g. the degree to which the available data are able to support the development of a process-driven model, even when the underlying processes are well understood
and can be described mathematically). This enables the models representing the various sub-processes to be developed by considering the degree of hypothetic and data influence that is most appropriate, thereby tailoring the model development process to the system under consideration. Given the conceptual nature of the framework, it provides high-level guidance and there is some subjectivity in its application to a particular case study. Details of the various components of the proposed framework are given in the following sub-sections.

**2.2 Identification of relevant sub-processes**

In order to apply the proposed approach, a suitable conceptual understanding of the processes that affect salinity in the system under consideration is required. In general, a distinction can be made between instream salt transport and the accession / addition of salt into the main channel via a variety of mechanisms that are affected by whether the river is operating under normal or flood conditions (Tefler et al., 2012), as illustrated in Fig. 2. The transport of instream salt occurs along the main
channel, from the top of the figure flowing through the river toward the bottom. Under normal conditions, the only potential

source of salt accession is generally via the inflow of saline groundwater, provided the stream is gaining (i.e. if the height of the water table is above the river level) (Fig. 3). However, during these conditions, salt from the groundwater store can also mobilise into various floodplain elements, such as wetlands and anabranches. In addition, the salt load in these floodplain elements increases as a result of evapo-concentration. Under flood conditions (Fig. 2), saline inflow from groundwater is likely to cease, as the river level is likely to be higher than the water table, resulting in a reverse of the flow direction (Fig. 3). Consequently, instead of the river gaining water (and salt) from groundwater, there will be a loss of fresh water from the river to recharge the groundwater system during a flood event.

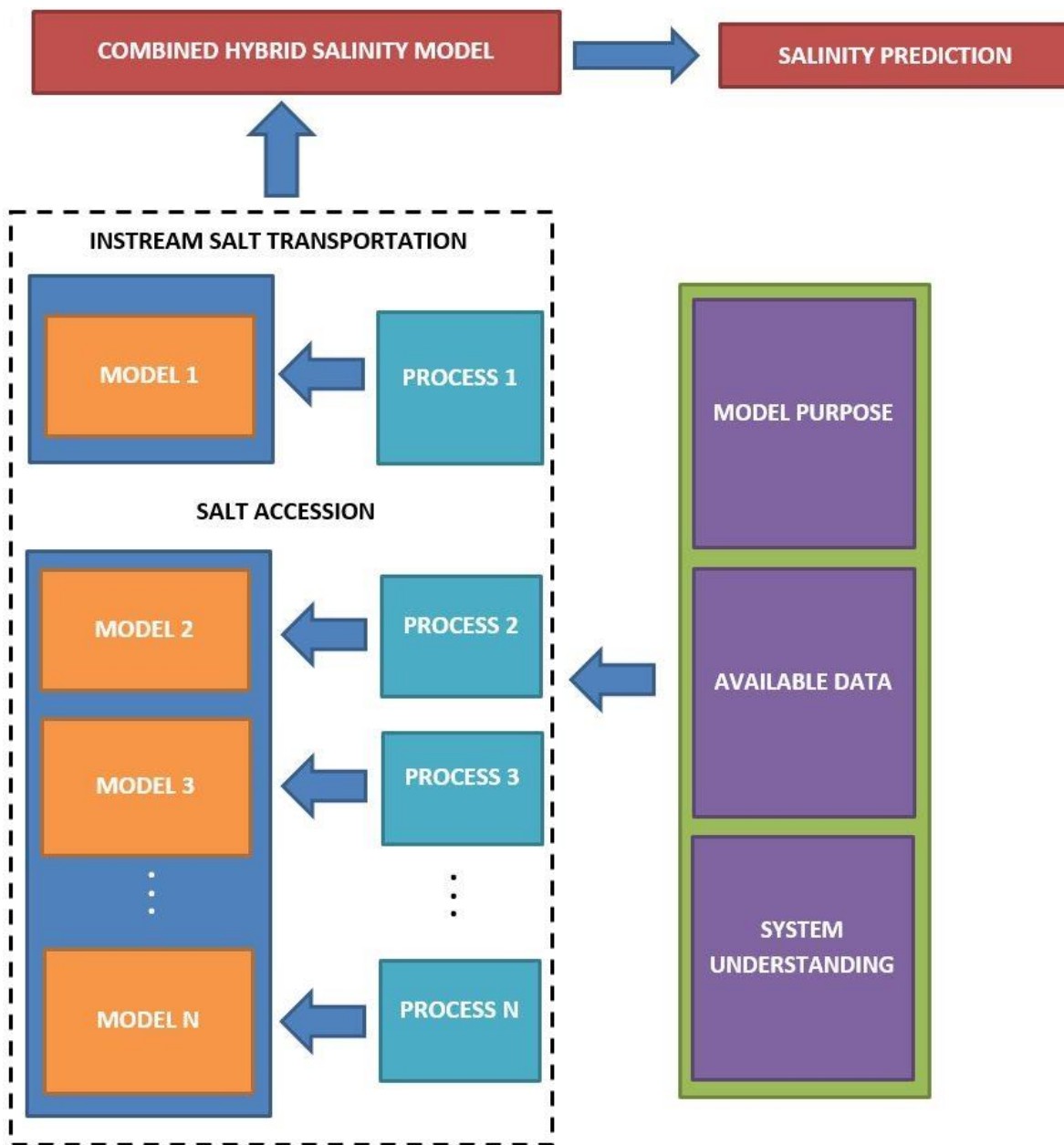

**Figure 1. Conceptualisation of proposed hybrid modelling approach.**

However, wetlands and anabranches that are disconnected under normal flow conditions may connect to the main river channel under flood conditions, adding the salt that has been building up in these systems since the last flood when water levels recede. The amount of salt added is a function of the magnitude of the flood and the time and conditions (e.g. degree of evapo-

concentration) since the occurrence of the last flood. Broader inundation of the floodplain results in recharge to the

groundwater system, and leads to increased flux, and hence salt load, from the groundwater system to the river once the river level returns to normal conditions. As part of the proposed approach, all of the specific sub-processes that contribute to salinity for the case study under consideration need to be identified.

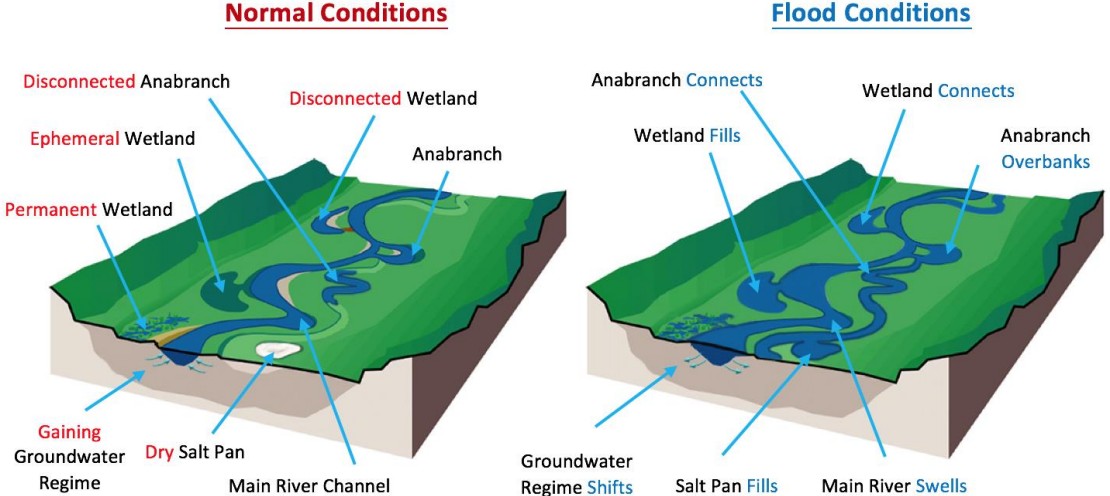

Figure 2. Processes affecting saline accessions during normal and flood conditions.


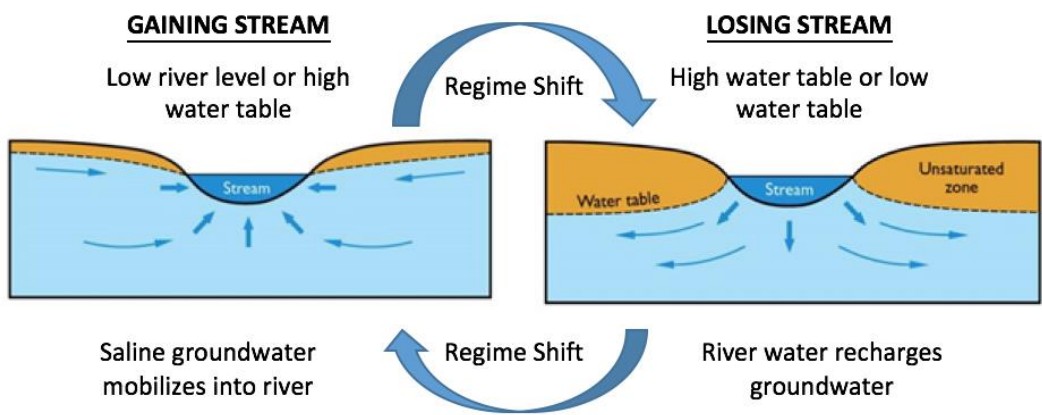

Figure 3. Processes affecting saline groundwater accessions for gaining and losing streams.


## 2.3 Identification of the most suitable model types

As part of this step, the model types that are most suitable for modelling the relevant sub-processes identified in the previous step are determined based on joint consideration of model purpose, system understanding and data availability and suitability (Fig. 1). Model purpose has an influence on which of the relevant processes have to be modelled explicitly. For example, if the overall purpose of the modelling exercise is to obtain salinity forecasts at downstream locations, there might not be a need to model all contributing processes explicitly (Maier and Dandy, 1996). In contrast, if the purpose of the model is to gain increased system understanding or to enable the impacts of various salt management options to be considered, all sub-processes will most likely have to be modelled explicitly. This also has an impact on which potential model inputs are considered. For example, if forecasting is the primary model purpose, auto-regressive values of the model output should be considered as potential inputs (e.g. Bowden et al., 2005b), as this is likely to improve the quality of the forecasts. In contrast, if the purpose of the model is to assess the impact of different management options on salinity, autoregressive values of the model output cannot be considered as potential model inputs, as the model output has to be independent of the model input(s) in such cases.

Once the processes for which sub-models are required have been identified, the most appropriate model type has to be selected for each of these. As mentioned previously, this requires an appropriate balance between hypothetic and data influence. The degree to which the selected processes are understood and can be described mathematically can be highly variable, as can the state of the available data to support different modelling approaches. For example, the transport of a conservative constituent with discharge, i.e. the process of instream salt transport, is generally well understood and requires relatively little data to be modelled explicitly, as the main processes consist of flow routing and storage. Consequently, the use of process models might be most appropriate. However, the same is unlikely to be true when modelling different processes of salt accession, as these are generally more complex, site specific (depending on soil types and groundwater conditions) and less well understood, making more hypothetically-influenced models an attractive alternative. If sufficient data are available, the use of universal function approximators, such as artificial neural networks (ANNs), might be best (Mount et al., 2016). However, a scarcity of data representing rare events, such as the large flood events that might be required to flush the salt stores in the wetlands and anabranches adjacent to the main river channel, might make models with a smaller number of parameters, such as regression, a better option.

It is important to note that the proposed framework is conceptual in nature and designed to provide high-level guidance. Consequently, its implementation for particular case studies is subjective. For example, how much data is required to support a particular modelling approach is case study dependent and relies on the judgement of the model developer. Consequently, this stage of the process may be iterative. Following the application of the developed sub-models, results may assist in identifying limitations in understanding or missing information. Hence, understanding gained from of the application of the

sub-model combinations developed may be applied to optimise the number and type(s) of sub-models included in the final hybrid model.

### 2.4 Development of required sub-models and hybrid model

Once the most appropriate model types have been determined for each of the sub-processes to be modelled explicitly, the corresponding models have to be developed. This process should follow state-of-the-art approaches for the development of the different types of models (see Jakeman et al., 2006; Maier et al., 2010; Wu et al., 2014; Hamilton et al., 2015; Galelli et al., 2014; Humphrey et al., 2017). Finally, the various sub-models have to be combined to develop the desired hybrid model (Kelly et al., 2013).

## 3 Case study

### 3.1 Background

In order to illustrate and test the utility of the conceptual high-level framework introduced in Section 2, it is applied to a case study of the 46 km reach between Lock 5 and Lock 4 on the River Murray in South Australia (Fig. 4). This particular location is chosen because:

- The reach is underlain by highly saline regional groundwater systems that provide significant salt accession into the river along this reach. As such, there are substantial saline accessions in this reach that are not well understood.
- This reach exemplifies a range of processes that are known to facilitate salt accession into a reach of a river, such as groundwater gain, inflow from streams, creeks and wetlands, and in-channel transport.
- Conditions along this reach have been monitored for many years, so there are suitable datasets for model development.
- The reach has had minimal changes in salinity management over time, so external influences on the underlying processes represented by the historical data are minimal.
- A number of changes are occurring in the River Murray system that will results in changes in the flow and inundation regime, and interest in the salinity response to these changes:
  - The Murray Darling Basin Plan, as per parts 1A and 2 of the Commonwealth *Water Act (2007)*, will return some water previously allocated to consumptive use to the environment, with the aim of increasing ecosystem health through processes such as increased frequency of inundation. The ability to predict the effects on salinity resulting from these changes is currently very limited.
  - Large-scale floodplain regulators are proposed in this reach to further increase inundation frequency to improve ecosystem health. A greater understanding of, and the ability to predict, salt accession under different conditions along this reach will improve the ability to manage and operate this infrastructure once constructed.

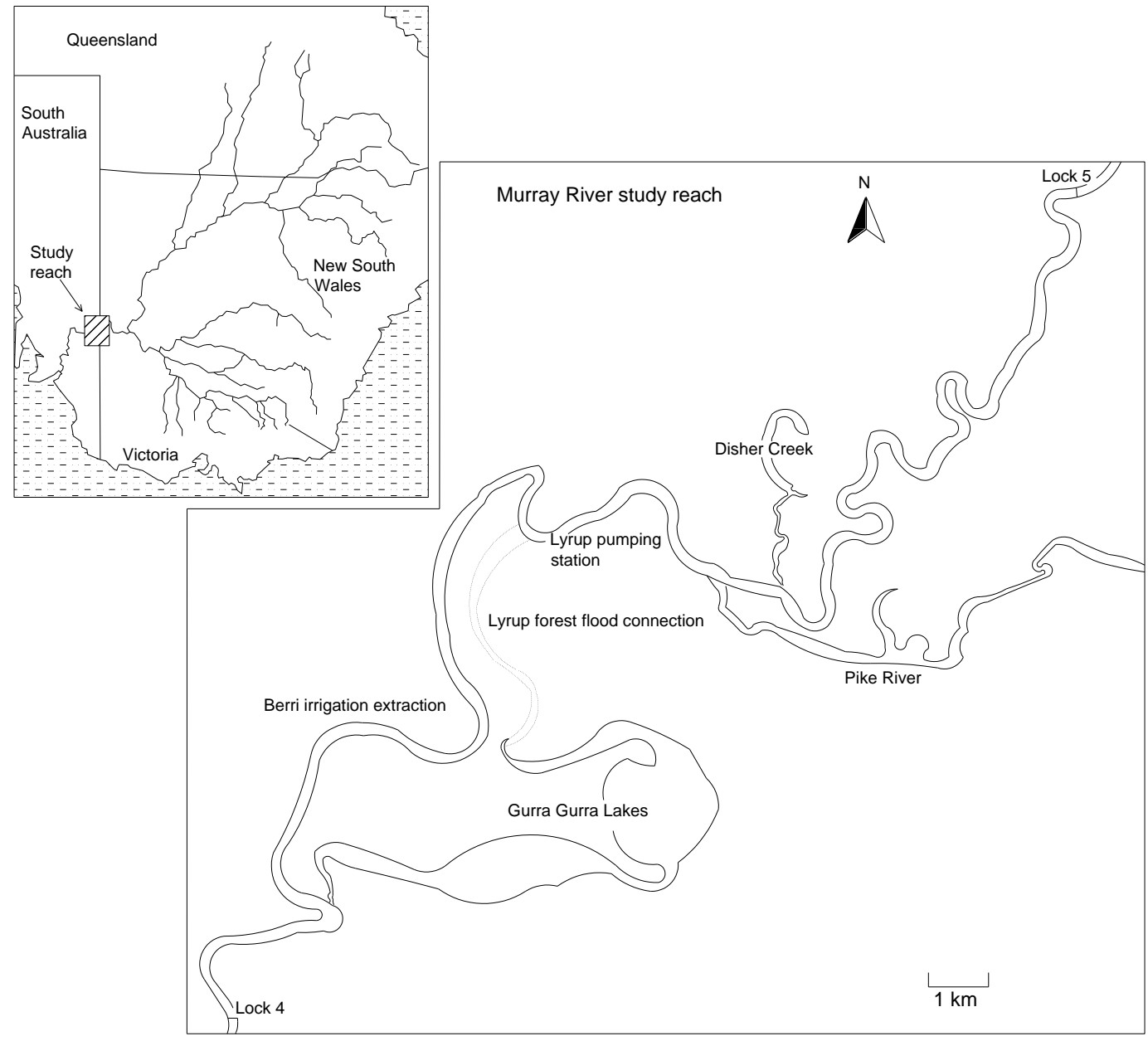

250

**Figure 4. Map of case study reach.**

## 3.2 Identification of relevant sub-processes

The main processes affecting salinity in the reach of interest include in-channel salt transport, as well as accession of salt via
groundwater and flushing of several large floodplains and wetlands.  There are also several anabranches and backwaters, which
include Pike River, the Gurra Gurra Lakes and Disher Creek (Fig. 4).  In-channel salt transport is primarily driven by advection
with flow in the main river channel, as well as the storage mixing volume behind the lock, which may change as water levels
are manipulated.

Saline accession via groundwater is a function of the relative levels of the groundwater adjacent to the river channel and the
water levels in the main river channel.  As discussed in Section 2.2, if the water levels in the main river channel are below the
adjacent groundwater levels, groundwater flows into the main river.  As the salinity of the groundwater that flows into the
main river channel in the reach of interest is very high (up to 70,000 mg $L^{-1}$) (Barnett, 2007), this can be a significant source
of saline accession.  Analysis of the available data from groundwater wells (WaterConnect, 2016) showed that groundwater
levels are relatively constant and that saline groundwater accession occurs at flows below approximately 40,000 ML $day^{-1}$,
after which overbank flow commences.  This is useful in that it demonstrates a link between groundwater accession and river
flowrate, and implies that models of such processes are unhelpful at higher flows.

There are three large salt sources that connect to the river at different flow rates: Disher Creek, Gurra Gurra Lakes, and the
Pike River.  Disher Creek is an evaporation basin for irrigation runoff and salt interception scheme flows.  Water is held in the
basin to be disposed of through evaporation, or if this is not sufficient, water can be pumped to the Noora Drainage Disposal
basin 20 km away from the river.  The management rules for Disher Creek allow for release of water from the evaporation
basin to the river at flows greater than 15,000 ML $day^{-1}$, which is assumed to be sufficient to dilute the highly saline releases
from the basin.  It should be noted that releases are not always triggered at this flow, and the release rate can be modified
depending on the measured salinity in the river at the time.

At lower river flows, the Gurra Gurra lakes are a terminal wetland system, with one connection to the river.  Under these
conditions, evaporation removes water from the wetland, which is then replaced from the river, and this process results in
naturally higher salinities than in the main river channel.  At flows of approximately 30,000 ML $day^{-1}$ the water level in the
river rises high enough to connect the flow path to the north of the lakes (through Lyrup forest), resulting in a through-flowing
system.  When this is the case, the lakes are flushed, and the saline water from the wetlands contributes to the main river
channel, thus increasing its salt content.

The Pike River is an anabranch that loops around Lock 5 and the upper area of the reach.  The lock provides a 3m head
difference, and due to this, the Pike River can flow around the lock and back into the river below it.  There are two inlets to

the Pike River above Lock 5, which are both regulated, historically to supply flows for irrigation purposes. At flows approaching 50,000 ML day$^{-1}$, the main river channel begins to overflow, and connects a number of temporary flow paths to the Pike River, resulting in wash-off and transport of salt that may have been deposited on the floodplain in that locality. This flow of 50,000 ML day$^{-1}$ is representative of when overbank flows start to occur along the reach between Locks 4 and 5 (and the Lower River Murray more broadly), where this process occurs along the river, as well as the longer-term process of recharge to groundwater and increased flux once river levels recede.

It should be noted that the amount of salt flushed into the main river channel from these systems is difficult to predict, as it is not only a function of flow, but also of the time between flushing events, the duration of an event and the nature of the flow regime at the time. Moreover, the accession processes are very different at different flow regimes. For example, at 15,000 ML day$^{-1}$, the salt contribution from Disher Creek may be represented by a point load into the main channel. However, at flows greater than 40,000 ML day$^{-1}$, Disher Creek disappears entirely from the map, as it is swallowed by an extensive floodplain. When this occurs, the creek does not contribute any additional salt to the system: it has already been entirely flushed and the only water in the area is flowing downstream as part of the flooded main river channel.

### 3.3 Identification of relevant sub-models

The purpose of the hybrid model is to quantify salinity responses to proposed managerial changes to the flow and inundation regime in the reach of the river under consideration under the Murray Darling Basin Plan. These changes will be enacted by the construction of additional control structures, and by selective releases or routing of volumetric flow. The salinity response to such changes is of interest in that the consequences of poor water quality can be high, and the modelling of different processes of accession is poorly understood. There is therefore value in identifying and modelling the main processes of accession separately, so that future management may determine the best locations of control options in addition to assessing the magnitude of their effects on salinity.

The data available for model development are shown in Table 1, including flow, temperature, stage height and salinity, which are measured at various points along the reach of interest. All historical data are available as daily readings. The recording and management of these data are undertaken by the South Australian Department of Water, Environmental and Natural Resources (DEWNR), largely funded through the Murray Darling Basin Authority (MDBA). The time periods for which data are available vary, depending on when a measuring station was constructed, and when the instruments to measure certain parameters were commissioned. Some datasets are deemed not suitable for model development and therefore excluded from this study, often due to short data records of a few years in length (e.g. measurements taken at the mouth of Gurra Gurra Lakes). Key locations include the two locks that define the extent of the reach considered, as well as the Pike River anabranch.

Berri river extraction and Lyrup pumping station, which are downstream of the Pike River, also provide useful information for salt transport and accession along the reach.

**Table 1. Details of available model data. All data are recorded at a daily resolution and have been sourced from DEWNR (Department of Environment, Water and Natural Resources).**

| Parameter | Location (Station Number) | Time Period |
|---|---|---|
| Flowrate (ML day$^{-1}$) | Lock 5 (downstream, A4260513) | 23 Jan 1981 – 25 May 2016 |
| Flowrate (ML day$^{-1}$) | Lock 4 (downstream, A260515) | 01 Jul 1983 – 30 Jun 2012 |
| Flowrate (ML day$^{-1}$) | Lyrup pumping station (A4260663) | 12 Nov 1993 – 18 Jun 2017 |
| Temperature (°C) | Berri irrigation extraction (A4260537) | 29 Mar 2001 – 02 Aug 2016 |
| Temperature (°C) | Pike River outlet (downstream, A260645) | 26 Sep 1991 – 18 Jun 2017 |
| Salinity (mg L$^{-1}$) | Lock 5 (upstream, A4260512) | 04 Jul 1972 – 01 May 2013 |
| Salinity (mg L$^{-1}$) | Lock 4 (upstream, A1260514) | 18 Jan 1994 – 20 Mar 2017 |
| Salinity (mg L$^{-1}$) | Pike River outlet (downstream, A260645) | 26 Sep 1991 – 18 Jun 2017 |
| Salinity (mg L$^{-1}$) | Berri irrigation extraction (A4260537) | 17 Oct 1942 – 20 Mar 2017 |
| Water level (m) | Lock 5 (downstream, A4260513) | 01 Apr 1924 – 01 May 2013 |
| Water level (m) | Lyrup pumping station (A4260663) | 11 Nov 1993 – 18 Jun 2017 |
| Water level (m) | Lock 4 (upstream, A1260514) | 01 Apr 1927 – 01 May 2013 |
| Water level (m) | Berri irrigation extraction (A4260537) | 01 Jan 1974 – 20 Mar 2017 |
| Salt load (kg day$^{-1}$) | Lock 4 (downstream, A260515) | 01 Jul 1983 – 30 Jun 2012 |

As discussed in Section 2.3, which modelling approach is most suitable for a particular process is a combination of the degree of process understanding and data availability. The relative degree with which these two factors are satisfied for the processes

to be modelled in this case study (see Section 3.2), based on a subjective assessment of available information, is summarised in Table 2. As can be seen, the processes affecting instream salt transport are well understood, able to be represented mathematically and supported by sufficient data to enable a process-driven model to be developed. However, the processes associated with the various modes of saline accession are considered to be not well understood, making data-driven models the best option. In relation to groundwater accession, the degree of available data is high, as this occurs during non-flood

events and relevant data are measured daily. Consequently, an artificial neural network is considered the most appropriate modelling approach due to its universal function approximation ability and its successful application to the prediction of salinity in the River Murray in previous studies (e.g. Maier and Dandy, 1996). However, as the saline accessions corresponding to overbank flow and flushing only occur during flood events, which occur infrequently, the data available on these processes is considered insufficient to support the development of a model with a potentially large number of parameters, such as an

artificial neural network. Instead, a linear regression model is considered most appropriate to represent these processes due to the combination of low degree of process understanding and low degree of data availability.

Table 2. Identified processes and the degree of data and understanding that are available for each.

| Process | Degree of Data Availability | Degree of Process Understanding | Selected Model Type |
|---|---|---|---|
| Instream salt transport | Medium | High | Process-driven |
| Overbank flow & flushing of Pike River | Low | Low | Linear regression |
| Overbank flow & flushing of Gurra Gurra Lakes | Low | Low | Linear regression |
| Overbank flow & flushing of Disher Creek | Low | Low | Linear regression |
| Groundwater accession | High | Low | Artificial neural network |

A conceptual representation of the resulting hybrid model is given in Fig. 5. As can be seen, the models corresponding to the four main sub-processes associated with saline accession identified in Section 3.2 are conceptualised as being applicable at different flowrates, which are determined based on preliminary analysis of the available data, with one model being applied to only one discrete bracket of flow. As a result, while each model primarily represents the process with which it is associated, it might also represent other processes that occur during the range of flows for which each model is developed. As shown in Fig. 5, the four models of saline accession (Models 2 to 5) feed into the instream salt transport model (Model 1).

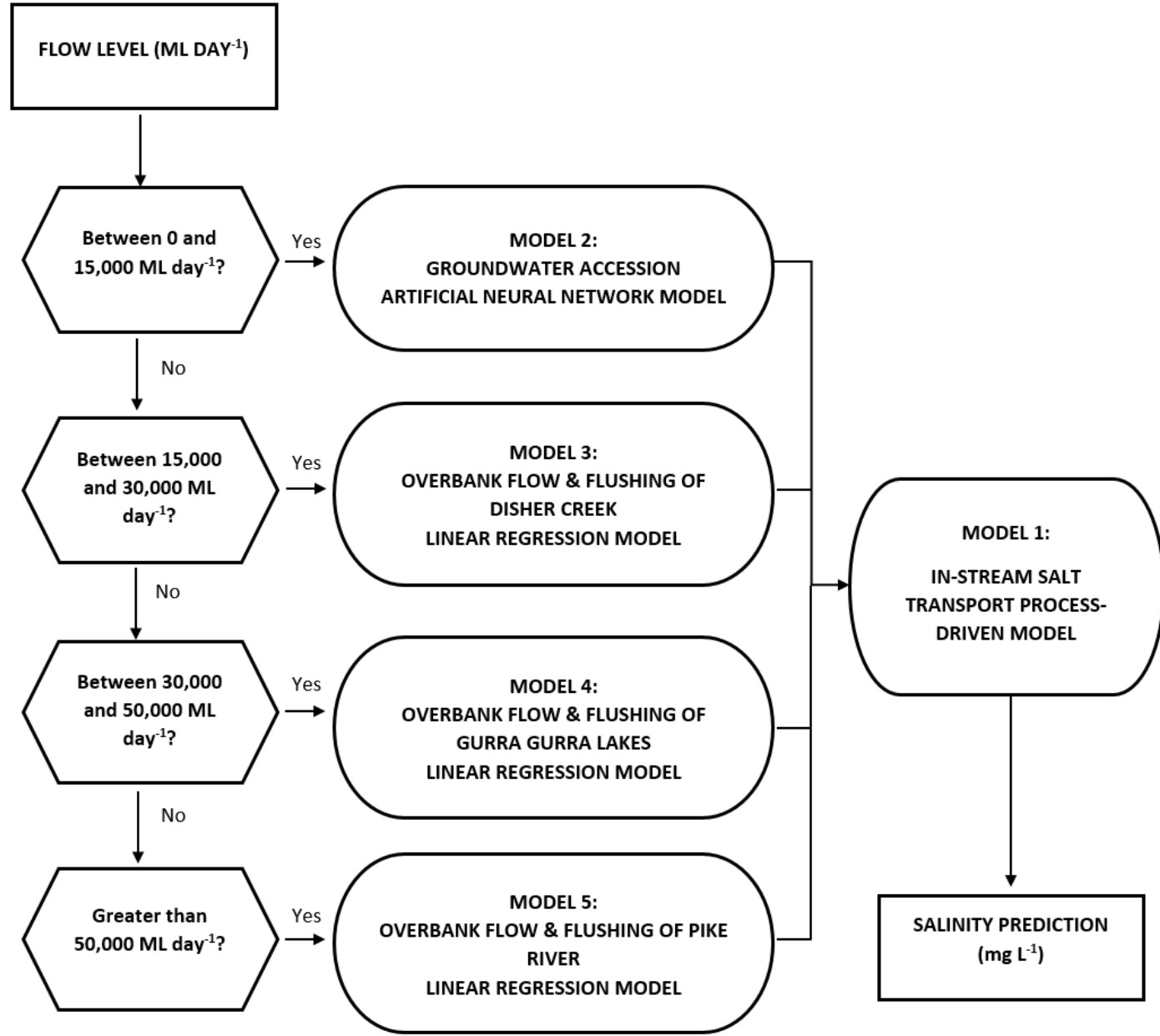

**Figure 5. Conceptual representation of components of hybrid model and how they are connected.**

### 3.5 Development of required sub-models and hybrid model

In this section, details of the development of the five sub-models (Fig. 5) are given. The modelling data and performance metrics are described first, as they are common to all models, followed by details on the development of the three different

modelling types (i.e. process-driven (Model 1), artificial neural network (Model 2) and linear regression (Models 3 to 5)). This is followed by details of the model development process for each of these model types, as well as how they are combined to form the hybrid model.


### 3.5.1 Model development data

The quality of the available data (see Section 3.1) is checked by visual inspection and a number of small periods of missing data (1-3 days) are filled in using linear interpolation, which is common practice for gaps in data of this duration and is unlikely to result in any significant loss of information (Kornelson and Coulibaly, 2014). Two longer periods of missing flow data at

Lock 5 in the period 2011-2012 (132 and 72 days, respectively) are filled in by correlation with corresponding water level data. To ensure consistency between sub-models, the longest common period of available data is used for the development of all models, which is from 18 Jan 1994 to 30 Jun 2012. The available data are split so that the first 80% (i.e. from 18 Jan 1994 to 21 Oct 2008) are used for model calibration and the subsequent 20% (i.e. from 22 Oct 2008 to 30 Jun 2012) are used for validation for all models. It should be noted that a regional drought event from 2001 to 2010, which is reflected in almost a

decade of low flows ($< 15,000$ ML day$^{-1}$), is the most significant unusual feature in the data and is purposely split between both calibration and validation datasets. To ensure all inputs into the ANN and regression models span the same ranges and can thus be combined during the modelling process, all data are standardised to have a mean of 0 and a standard deviation of 1, as per Eq. (1).

$$y = \frac{(x - \bar{x})}{\sigma} \tag{1}$$

- $y$ = standardised input (dimensionless).
- $x$ = original input (with units).
- $\bar{x}$ = mean of original inputs.
- $\sigma$ = standard deviation of original inputs.

**3.5.2 Model performance assessment**

All data are calibrated and validated against the salinity at Lock 4. The root-mean-squared error (RMSE, Eq. (2)) and the Nash-Sutcliffe Efficiency (NSE, Eq. (3)) are used as metrics to judge the fit of the predicted variables to the observed data when calibrating the parameters of the models.

$$RMSE = \sqrt{\frac{1}{n} \sum_{i=1}^{n} (x_m^i - x_o^i)^2} \tag{2}$$

$$NSE = 1 - \frac{\sum_{i=1}^{n}(x_m^i - x_o^i)^2}{\sum_{i=1}^{n}(x_o^i - \bar{x}_o)^2} \tag{3}$$

- $n$ = number of points in the series.
- $x_m$ = modelled points.
- $x_o$ = observed points.
- $\bar{x}_o$ = mean of observed points.

The goodness of fit of the hybrid model is evaluated against two benchmark models, one process-driven and one data-driven, with the metric $G_{bench}$ introduced in Seibert, 2001 (Eq. 4).

$$G_{bench} = 1 - \frac{\sum_{i=1}^{n}(x_o^i - x_m^i)^2}{\sum_{i=1}^{n}(x_o^i - x_b^i)^2} \tag{4}$$

- $x_b$ = benchmark model data points.

The $G_{bench}$ index is structured similar to the NSE, but replaces the mean of the observed time series with the time series of a benchmark model, so an index of zero indicates that model performance is equal to that of the benchmark model, with negative values indicating that the performance of the model under consideration is inferior to that of the benchmark model and positive values indicating the opposite

### 3.5.3 Process-driven salt transport sub-model (Model 1)

The purpose of this model is to simulate the instream transport of salt from Lock 5 to Lock 4, thereby predicting salinity at Lock 4 as a function of the upstream salt load, without considering saline accessions due to groundwater inflow or the flushing of anabranches and backwaters along the reach. The model is developed using eWater Source (Welsh et al., 2013). Routing is represented using a piecewise linear lookup table, where the travel times for key flow rates are calculated based on travel times of flow peaks in the historical record. A dead storage volume is used to represent the mixing time for salinity, as the travel time for solutes is much slower than the wave celerity travel time resulting from the analysis of flow peaks. The routing or transportation method is a fully mixed water quality constituent. This technique has been used since the 1970s in various water quality models and hence is well developed. The key principles are that the mass balance of the modelled constituents (e.g. salt) is maintained in all divisions of all links (which represent a reach). Calculations take place for every time step, which is daily.

Development of the travel times and dead storage volumes for all areas of the river were calibrated in a previous study, as outlined in MDBC (2002). The salt constituent volumes in the upstream reaches were also determined as part of this previous work, however, the salt accession within the study reach itself is considered as part of the calibration process in this study. As

this is a process-driven model, the required inputs are pre-determined due to the mathematical specification of the model. These include upstream flow and salinity at Lock 5, and knowledge of the physical characteristics of the reach, such as its total length and the location of various extraction points (e.g. Lyrup pumping station and Berri irrigation extraction). The transport model routes the upstream salt through the reach down to Lock 4, and converts it into salinity by multiplying the salt load with

the rate of flow. The difference between this transported salinity and the measured salinity at Lock 4 (i.e. the residual salinity of Model 1) represents the salt that is gained by the river due to the accession processes that occur within the reach itself (Section 3.2), and is the salinity that is predicted by the remaining hybrid component models (i.e. Models 2, 3, 4 & 5).

### 3.5.4 Artificial neural network (ANN) groundwater accession model (Model 2)

The purpose of this model is to predict accession of salt for flows that are less than 15,000 ML day$^{-1}$, which is sourced primarily

from groundwater inflows and small increases in water level at the upstream end of the reach between locks 5 and 4. These tend to occur due to the same mechanisms along large stretches of the river during a single dry or wet event. However, their response times can be vastly different, ranging from days for responses to changing water levels, to months for the slower responses to saline groundwater accessions.

The model is implemented using the Validann R-package (Humphrey et al., 2017) using the steps in the ANN model development process outlined in Maier et al. (2010) and Wu et al. (2014). Potential input variables considered include salinity, temperature, water level and flowrates, measured at various time lags and locations along the length of the reach (Table 1). It should be noted that past values of the model output are not considered as potential model inputs, as the purpose of the model is to assess the impact of different management options on river salinity, rather than forecasting salinity future salinity values

(see Section 2.3). The model output is the residual salinity between the process-driven in-channel salt transport model (Model 1) and the measured salinity at Lock 4, for the days for which the measured flow at Lock 5 is less than 15,000 ML day$^{-1}$. Appropriate model inputs are determined with the aid of correlation analyses between potential model inputs and the model output, resulting in the selection of two inputs, including salinity at Lock 5 and the water level measured at Lyrup pumping station (Fig. 6), which are lagged by five and three days, respectively. The selected inputs reflect that saline accessions at low

flows are primarily driven by upstream salinities (primarily affecting water level) and water levels (primarily affecting groundwater inflow).

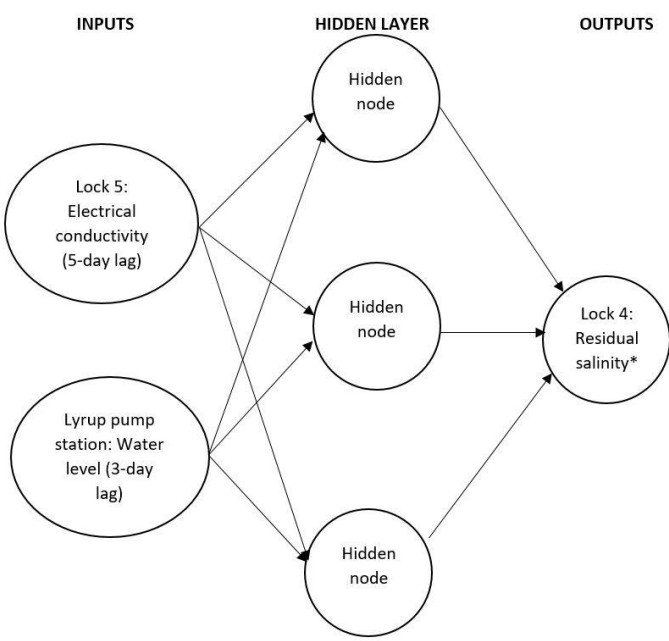

**Figure 6. Structure and inputs of ANN model for predicting saline groundwater accession as part of the hybrid model (Model 2).**
*The residual salinity output is the difference between the measured salinity and the output from the process-driven salt transport model
for all flowrates that are less than 15,000 ML day$^{-1}$.

Multilayer perceptrons (MLPs) are used as the model architecture, as this is by far the most commonly used architecture in ANN applications in hydrology and water resources (Maier et al., 2010; Wu et al., 2014). Different combinations of standard activation functions (linear, sigmoidal and hyperbolic tangent) are trialled on the calibration data, resulting in the selection of the hyperbolic tangent function applied to the single hidden layer and the linear function being applied to the output layer. An ANN with three hidden nodes performs best on the calibration data, based on trials with one to four hidden nodes. The ANN is fully connected, so there are nine weighted connections (i.e. nine parameters to calibrate). The development data contain 5391 points (i.e. 5391 days over the calibration period), so there are almost 600 data points available to calibrate each parameter, making overfitting unlikely. These parameters are calibrated using the Broyden-Fletcher-Goldfarb-Shanno (BFGS) algorithm, as this method generally performs well when simulating hydrological phenomena (Zounemat-Kermani et al., 2016). Optimal values of the learning rate and momentum are obtained using trial and error on the calibration dataset, and the initial weights are selected randomly on the range [-0.5,0.5]. To test against overfitting, the model is validated against the residual salinity between the transport model (Model 1) and the measured salinity at Lock 4, over the period 22 Oct 2008 to 30 Jun 2012 for all data points corresponding to flowrates at Lock 5 of less than 15,000 ML day$^{-1}$.

### 3.5.5 Linear regression models for Disher Creek, Pike River and Gurra Gurra Lakes accession (Models 3 to 5)

The purpose of the linear regression models (Models 3 to 5, Fig. 5) is to predict accessions of salt to the river at flows in excess of 15,000 ML day$^{-1}$, which are dominated by Disher Creek and Pike River at a point downstream from Lock 5 (see Fig. 4), and the inflow from the Gurra Gurra Lakes at a point upstream from Lock 4 (Fig. 4), and are also likely to include saline accessions from groundwater and increases in water level at flows closer to 15,000 ML day$^{-1}$. The data in Table 1 are available as potential inputs for model development, with the most relevant inputs determined by trial and error during the calibration process. Because of the scarcity of peak flow data and the complexity of the physical processes being modelled, it is valuable to incorporate as much process understanding into these models as possible. This is achieved by ensuring that some of the more important known process drivers, such as peak duration, time since last peak flow and historical flow volumes, are represented as potential model inputs, as follows:

- The five-year historical volume of water (ML) is extracted from the daily flowrate measurements at Lock 5 ($Q_2$) because residual waters from receding historical peaks or overbank flows can create concentrated pockets of salt once the water has evaporated, which is then available to be accessed by the next overbank flow.

- The duration of the peak flow event (D) is extracted as a count of days, which begins incrementing at the commencement of a peak flow event, because longer floods allow the extended floodplain more time to connect with groundwater aquifers at a distance from the river, and to react with the salt content of the soil.

- The time since last peak flow (T) is extracted as a count of days, which begins incrementing when the daily flowrate falls below the defined peak flow, because a longer time since the last peak allows for a greater amount of saline groundwater to seep into depressions and shallow reservoirs that may be some distance from the main channel, thereby increasing the amount of salt available for overbank flow.

For the purpose of this work, a peak flow event begins when the flowrate at Lock 5 is larger than the lower flow bound of an individual model. For example, a peak flow event for Model 3 is any daily flow that is 15,000 ML day$^{-1}$ or higher, while a peak flow event for Model 5 is any daily flow that is 50,000 ML day$^{-1}$ or higher.

The models are developed in Microsoft Excel, using the Solver function (i.e. a gradient method) to optimise the coefficients from a range of starting positions to minimise the chance of identifying locally optimal parameter values. The models are calibrated against the residual salinity at Lock 4 obtained from the process-driven instream salt transport model (Model 1), over the time period from 18 Jan 1994 to 21 Oct 2008, using the NSE as the objective function. Data from 22 Oct 2008 to 30 Jun 2012 are used for validation, however, flow data from 18 Jan 1989 are also used in order to calculate the summed, volumetric flow from five years previous ($Q_2$), which is considered as a potential input for these models, as mentioned above.

The resulting equations for Models 3, 4, and 5 are given by Eqs. (5), (6) and (7), respectively.

**Model 3**

$$RS_p = 0.4EC + 0.35Q_2 + 0.1D, \qquad if \ 15,000 \leq Q_1 < 30,000 \tag{5}$$

**Model 4**

$$RS_p = 0.4WL + 0.3T + 0.35D, \qquad if \ 30,000 \leq Q_1 < 50,000 \tag{6}$$


**Model 5**

$$RS_p = 0.3WL - 0.45Q_1, \qquad if \ Q_1 > 50,000 \tag{7}$$

Where:

- $RS_p$ = Residuals between the measured salinity and the output salinity from the process-driven salt transport model for all peak flowrates greater than or equal to 15,000 ML day$^{-1}$ (mg L$^{-1}$).
   - $Q_1$ = Flow rate downstream of Lock 5 (ML day$^{-1}$).
   - $Q_2$ = Five year historical volume of water at Lock 5 (ML).
   - WL = Water level at Lyrup pump station (m).
- T = Time since last peak flow (days).
   - D = Duration of peak flow event (days).
   - EC = Salinity downstream from Lock 5 (mg L$^{-1}$).

The selected inputs for Model 3 indicate a positive correlation with salinity at Lock 5 (EC), the five year historical flow volume
at Lock 5 ($Q_2$) and peak flow event duration (D).  The positive correlation with EC is most likely related to shallow overbank flow, as discussed in Section 3.5.4.  $Q_2$ is likely to increase salt load, as greater volumes of historical flow indicate a greater likelihood that events that generate saline accessions have occurred in the previous five years, while D is likely to be positively correlated with saline accessions, as longer flood durations provide more time for groundwater recharge during an event, and hence later discharge on the recession of a flood, as slow response saline accessions.


The selected inputs for Model 4 indicate a positive correlation with higher water levels at Lyrup pump station (WL), time since last peak flow (T) and peak flow event duration (D).  The positive correlation with WL is most likely due to the fact Lyrup pump station is just upstream of the connection that flushes Gurra Gurra Lakes at high flows, with higher water levels at Lyrup pump station providing an indication of an increased ability to flush the lakes.  Higher values of time since last peak flow are
likely to increase saline accessions, as longer periods of time between floods provide more time for evapoconcentration to

occur, as well as saline groundwater to flow into the Gurra Gurra Lakes system, which is very shallow. Finally, as is the case for Model 3, longer flood durations provide more time for stored saline water to be flushed into the main river channel.

The selected inputs for Model 5 indicate a positive correlation with higher water levels at Lyrup pump station (WL) and a negative correlation with flow downstream of Lock 5 ($Q_1$). The positive correlation with WL is most likely related to the fact that larger areas of the floodplain are inundated at higher water levels, increasing overall salt load. The negative correlation with flow is most likely due to the increased dilution of the salt load at the high flows to which this model caters.

### 3.5.6 Hybrid model

A schematic of the resulting hybrid model is shown in Fig. 7. As can be seen, the process-driven instream salt transport model (Model 1) forms the basis of the hybrid model, with the different types of accessions, modelled using the ANN and regression models, added at appropriate locations. Specifically, the first and second regression models (Models 3 and 4, respectively) are added downstream of Lock 5, at the approximate location of the Pike River and Disher Creek outlets. The third regression model (Model 5) is added upstream of Lock 4, near the entrance of Gurra Gurra Lakes. The output from the groundwater accession ANN (Model 2) is added to Model 1 at Lock 4. Although groundwater accession occurs along the length of the reach under consideration, these inflows are impractical to segregate. The outputs from Models 2 to 5 are only added to Model 1 when triggered by the corresponding flowrate in a given timestep: there is only one model besides Model 1 that is describing the salinity levels on any given day.

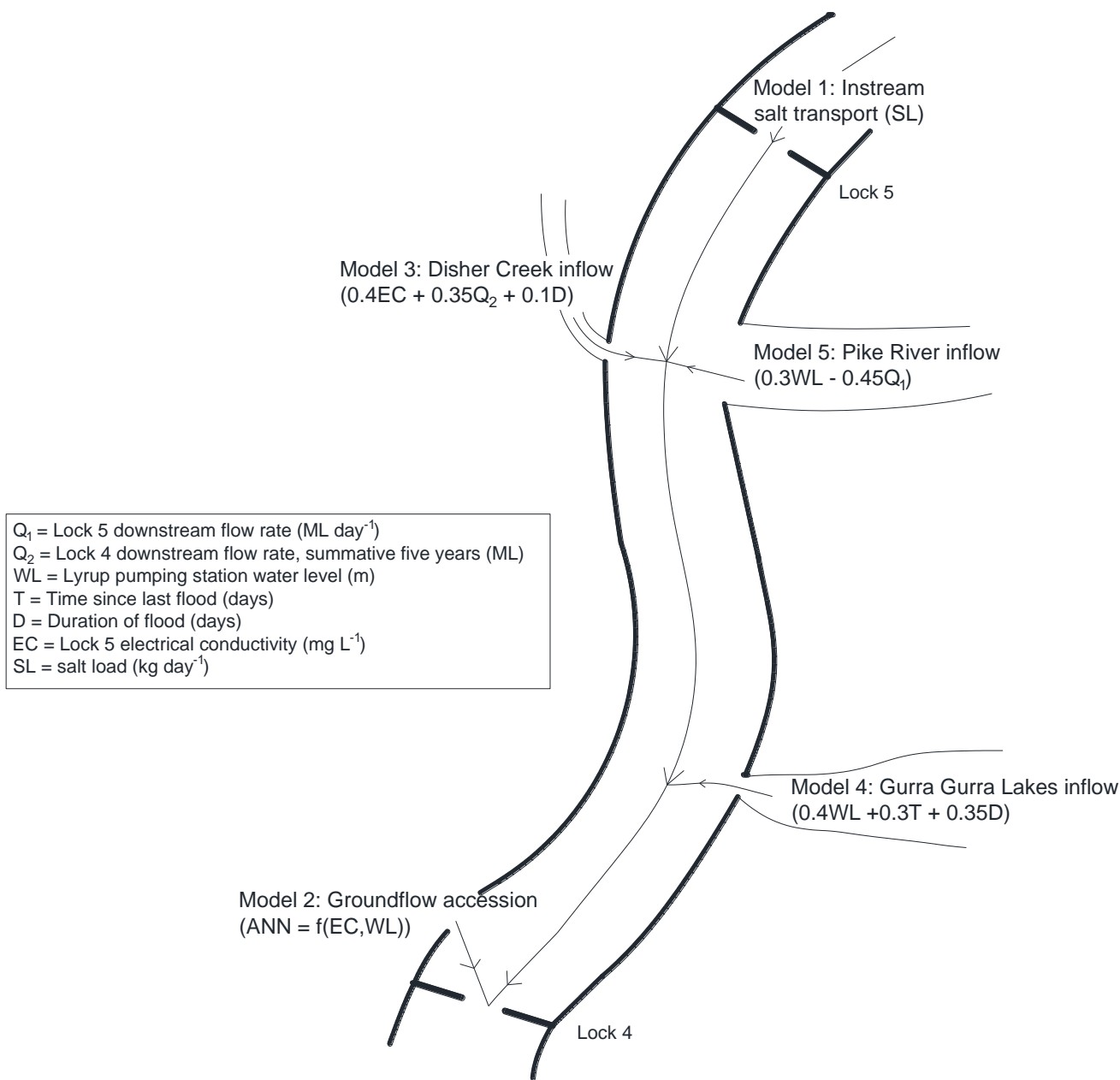

Model 1: Instream
salt transport (SL)

Lock 5

Model 3: Disher Creek inflow
$(0.4EC + 0.35Q_2 + 0.1D)$

Model 5: Pike River inflow
$(0.3WL - 0.45Q_1)$

$Q_1$ = Lock 5 downstream flow rate (ML day$^{-1}$)
$Q_2$ = Lock 4 downstream flow rate, summative five years (ML)
WL = Lyrup pumping station water level (m)
T = Time since last flood (days)
D = Duration of flood (days)
EC = Lock 5 electrical conductivity (mg L$^{-1}$)
SL = salt load (kg day$^{-1}$)

Model 4: Gurra Gurra Lakes inflow
$(0.4WL +0.3T + 0.35D)$

Model 2: Groundflow accession
(ANN = f(EC,WL))

Lock 4

**Figure 7. Schematic of the hybrid model on a stylised view of the river (not to scale). The regression models are input as point loads, while the groundwater accession is added to the transport model at Lock 4.**

### 3.6 Development of benchmark models

To enable the predictive ability of the hybrid model to be assessed in an objective manner, it is compared with that of two
benchmark models that represent commonly used paradigms for modelling salinity in rivers in previous studies, including

data-driven ANN and process-driven models. As summarised in Table 3, although the process-driven benchmark model does account for saline accession along the reach of interest explicitly, this is done via the addition of average historical accessions, as this represents the best available information. Consequently, unlike the hybrid model, which accounts for saline accession in a dynamic fashion, the benchmark process-driven model does so in static fashion. However, in-stream salt transport
processes are represented in an explicit and dynamic manner. In contrast, in the benchmark ANN model, all processes are represented implicitly, as it predicts salinity at Lock 4 as a function of available data along the length of the reach of interest, without explicit consideration of any of the underlying processes. However, this is done in a dynamic fashion. For the sake of consistency with the development of the hybrid model, the calibration and validation data, the model development processes and the way model performance is assessed are identical to those used for the development of Model 1 (process-driven
benchmark model) and Model 2 (ANN benchmark model). Further details of the development of the two benchmark models are given in the subsequent sections.

**Table 3. Method by which different processes are represented by the hybrid and benchmark data- and process-driven models. The inputs are represented either explicitly (by separate processes within the model) or implicitly. The outputs are either dynamic (the salt load varies in response to some time-dependent environmental changes) or static.**

| Process | Model | Data-driven benchmark model | | Process-driven benchmark model | | Hybrid model | |
|---|---|---|---|---|---|
| In-stream salt transport | implicit | dynamic | explicit | dynamic | explicit | dynamic |
| Groundwater accession | implicit | dynamic | explicit | static | explicit | dynamic |
| Pike River inflow | implicit | dynamic | explicit | static | explicit | dynamic |
| Gurra Gurra Lakes inflow | implicit | dynamic | explicit | static | explicit | dynamic |
| Disher Creek inflow | implicit | dynamic | explicit | static | explicit | dynamic |

### 3.6.1 ANN model (benchmark)

The purpose of this ANN model is to predict the total salinity in the river at Lock 4 directly, which in contrast to the ANN
model that forms part of the hybrid model (Model 2), which predicts the residual salinity between the process-driven instream salt transport model predictions and the measured salinity at Lock 4 for flows up to 15,000 ML day$^{-1}$, aimed at representing groundwater accession and low flow processes only. A mentioned above, the benchmark ANN model is developed using the same methodology as that used for Model 2 (see Section 3.5.4). A summary of the resulting ANN model is given in Fig. 8.

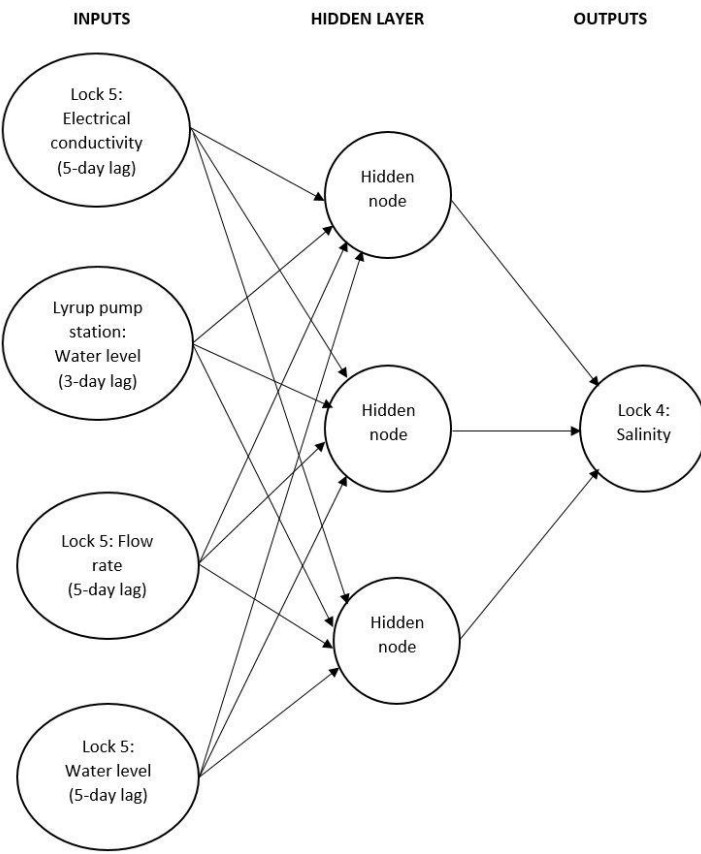

INPUTS     HIDDEN LAYER     OUTPUTS


**Figure 8. Structure and inputs of the benchmark ANN.**

### 3.6.2 Process-driven model (benchmark)

The benchmark process-driven model is identical to the one used in the hybrid model. However, as shown in Table 3 and described above, whereas salt accession within the study reach is modelled dynamically using ANN and regression models in the hybrid model, in the benchmark model, salt accession is represented as an average of historical salt loads. This is done by calculating the average daily salt load at Lock 4 for the calibration period of 18 Jan 1994 to 21 Oct 2008, which is then applied as two constant point loads. Most (approximately 82%) of this load is applied upstream of the Berri irrigation extraction, as this area forms a longer part of the study reach than the area downstream of this location. The remainder of the constant salt load is applied downstream from Berri.

## 4. Results and discussion

The time series plots of actual versus predicted salinities for the hybrid model and its component models for the validation data are given in Fig. 9, with the corresponding performance statistics given in Table 4. As can be seen, the hybrid model performs very well, with a NSE of 0.89 and an RMSE of 12.62 mg L$^{-1}$ (for data ranging from approximately 50 mg L$^{-1}$ to 250 mg L$^{-1}$). The time series plot shows that the model has captured all variations in salinity very well, with only small over- or under-predictions (Fig. 9c).

The performance of the process-based in-stream salt transport model (Model 1) on its own is not very good, with a NSE of -1.56 and an RMSE of 61.70 mg L$^{-1}$. This is because this model does not include any of the saline accessions within the reach, and therefore under-predicts salinity values significantly. However, the model is able to capture most major variations in salinity (Fig. 9a). When the ANN model designed to primarily capture saline groundwater accessions (Model 2) is added, the NSE increases to 0.38 and the RMSE reduces to 30. 46 mg L$^{-1}$, which is due to improved performance during periods of low flow, as expected (Figs. 9b, d). When the linear regression models designed to capture saline accessions from the flushing of Disher Creek, Gurra Gurra Lakes and Pike River (Models 3 to 5) are added, an NSE value of 0.89 and an RMSE value of 12.62 mg L$^{-1}$ are achieved as a result of increased performance during high flow periods. This indicates the value of the proposed hybrid approach, as each of the models of the different sub-processes improve model performance significantly.

The hybrid model also performs favourably compared with the two benchmark models, as shown in Table 5 and Fig. 10. The process-driven benchmark model performs significantly worse than the hybrid model, with NSE and RMSE values of -0.14 and 41.10 mg L$^{-1}$, respectively, compared with corresponding values of 0.89 and 12.62 mg L$^{-1}$ for the hybrid model. As can be seen in Fig. 10, this is due to an over-prediction of saline accessions by the benchmark process-driven model, as these are based on static historical values, rather than being modelled dynamically as a function of changes in flow, water levels and upstream salinity, as is the case for the hybrid model. However, addition of the average values of the historical saline accessions results in an improvement in model performance compared with that of the process-driven in-stream salt transport model used in the hybrid model (Model 1), with an increase in NSE from -1.56 to -0.14 and a reduction in RMSE from 61.70 mg L$^{-1}$ to 41.10 mg L$^{-1}$.

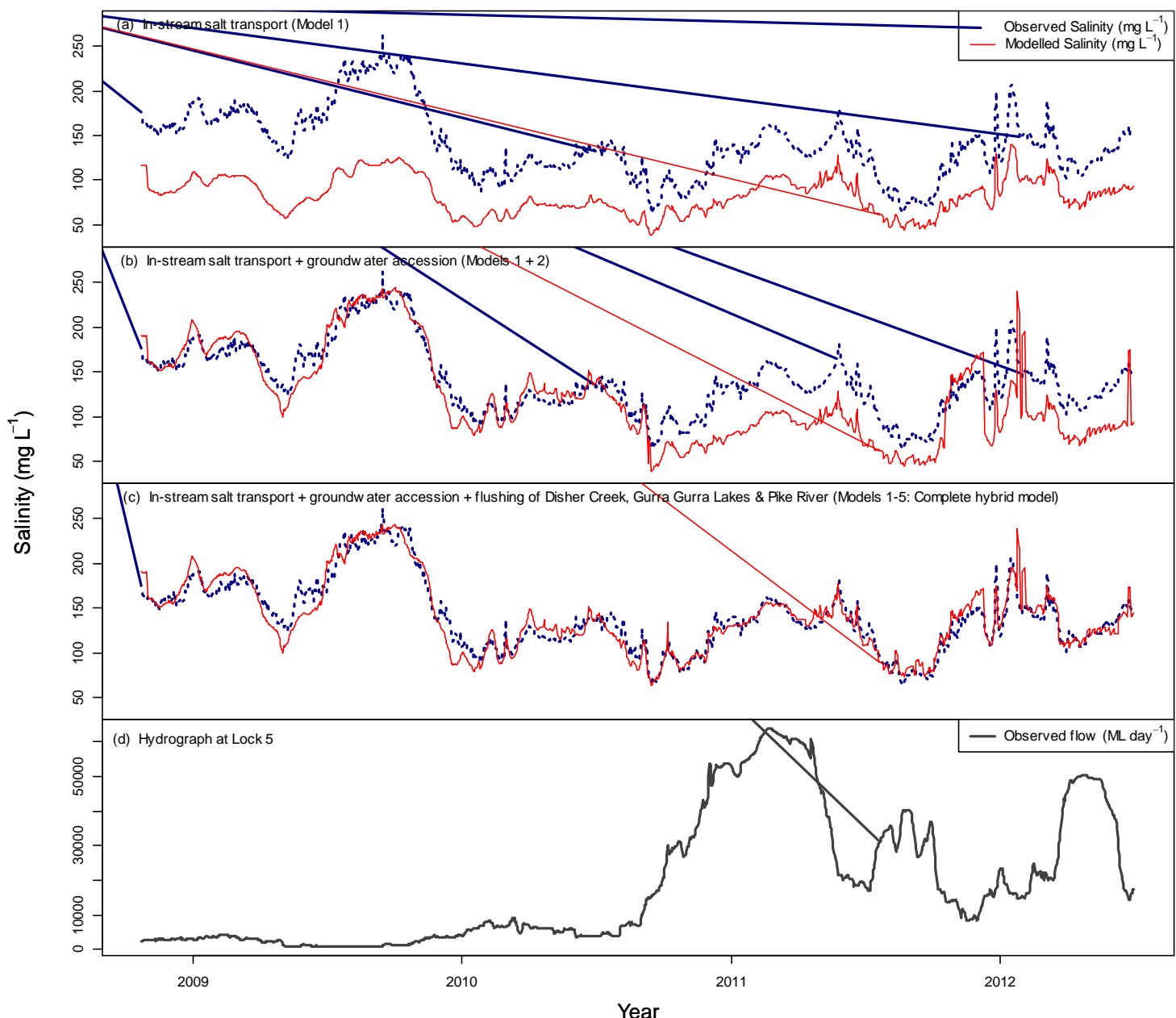

**Figure 9. Measured versus modelled salinity at Lock 4 for the hybrid model and its components for the validation data, as well as corresponding hydrograph for Lock 5.**

**Table 4. Performance statistics for the hybrid model and its component models for the validation data.**

| Model | NSE | RMSE (mg L$^{-1}$) |
|---|---|---|
| In-stream salt transport (Model 1) | -1.56 | 61.70 |
| In-stream salt transport + groundwater accession (Models 1 + 2) | 0.38 | 30.46 |
| In-stream salt transport + groundwater accession + flushing of Disher Creek, Gurra Gurra Lakes & Pile River (Models 1-5 – complete hybrid model) | 0.89 | 12.62 |

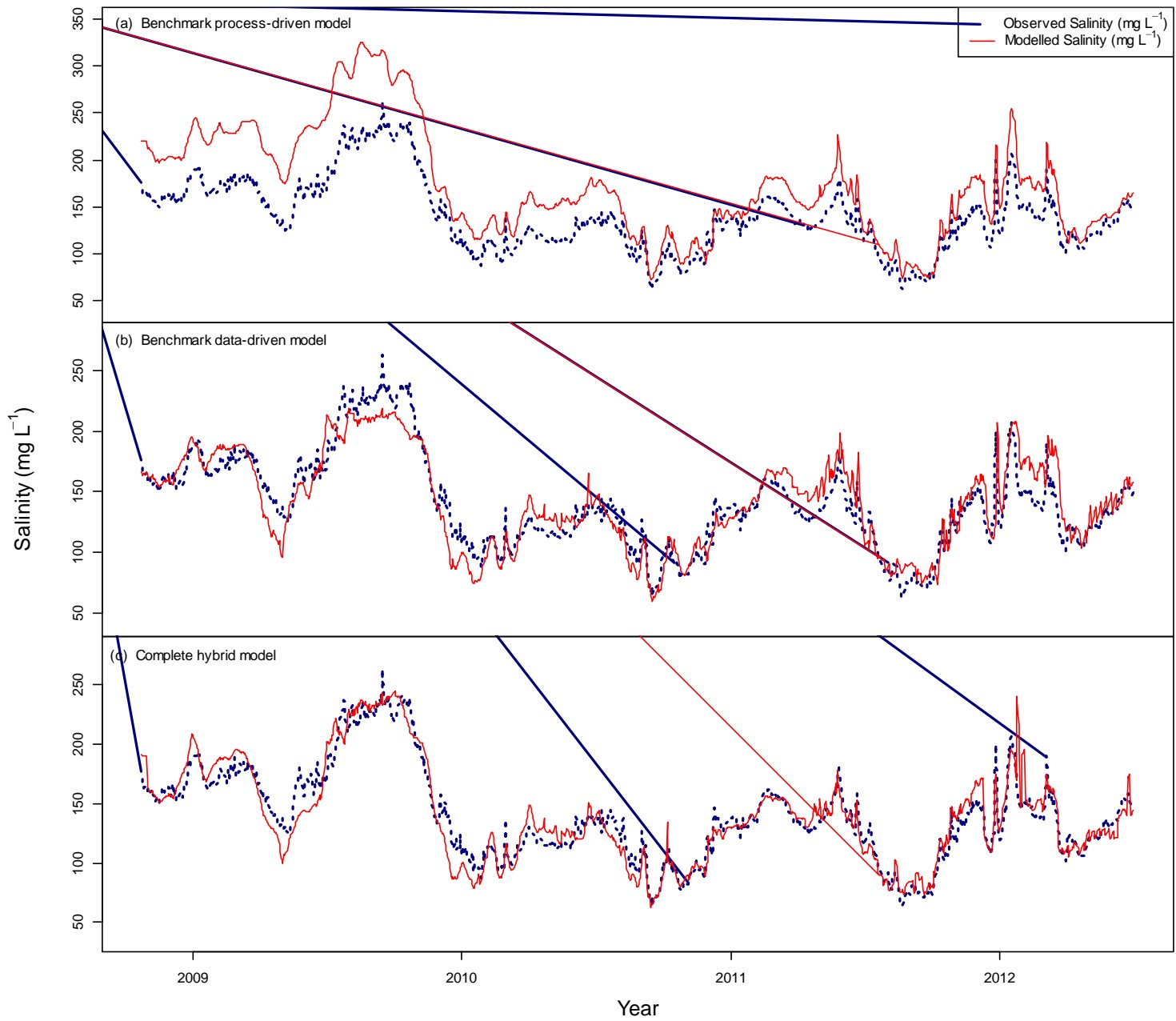

**Figure 10. Measured versus modelled salinity at Lock 4 for the hybrid model and the two benchmark models for the validation data.**

**Table 5. Performance statistics for the two benchmark models and the hybrid model for the validation data.**

| Model | NSE | RMSE (mg L$^{-1}$) |
|---|---|---|
| Process-driven benchmark model | -0.14 | 41.10 |
| Data-driven benchmark model | 0.83 | 15.93 |
| Hybrid model | 0.89 | 13.58 |

In contrast to the process-driven benchmark model, the data-driven benchmark model performs only slightly worse than the hybrid model, with NSE and RMSE values of 0.83 and 15.93 mg L$^{-1}$, respectively, compared with corresponding values of 0.89 and 13.58 mg L$^{-1}$ for the hybrid model. As can be seen in Fig. 10, the primary differences between the hybrid and data-driven benchmark models are that the benchmark model under-predicts saline accessions during low-flow periods (e.g. towards the end of 2009) and over-predicts saline accessions during high-flow periods (e.g. in the first half of 2011 and 2012). This highlights the benefits of the hybrid model in being able to tailor models to accessions during low- and high-flow periods. The superior performance of the hybrid model is reinforced by positive values of the $G_{bench}$ index of 0.36 and 0.90 for the data-driven and process-driven benchmark models, respectively. .

In addition to resulting in improved predictive performance, a major benefit of the hybrid model is that it, unlike both benchmark models, can be used to assist with dealing with some of the proposed changes in river management, in part driven by the Murray Darling Basin Plan. Historically, river management has focused on ensuring supply of water for consumptive demands. However, more recently, river management is being expanded to include the improvement of environmental outcomes. This includes changing the flow regime with the delivery of environmental water, and by the construction of control infrastructure on the floodplain to increase inundation frequency and duration. As the variables that affect the flow regime are included as inputs in the hybrid model, this model can be used to assess the impact of some of the proposed management options on river salinity. In addition, the hybrid model contributes to an increased understanding of the underlying processes.

Overall, the results of the illustrative case study highlight the potential benefits of the proposed framework. By considering the relevant processes affecting river salinity at the site of interest, how well they are understood and can be represented mathematically, how much data there is to support model development and what the primary purpose of the model is, a hybrid model was able to be developed that not only results in better predictive performance than the corresponding benchmark process- and data-driven models, but is also more useful from a management perspective. However, given the conceptual nature of the proposed framework and the level of subjectivity required to implement it, it is not possible to tell if an even better model could have been developed had different decisions been made with regard to model types.

## 5. Summary and conclusions

This paper introduces a framework for the development of hybrid models for the prediction of salinity in rivers. As part of the framework, relevant sub-processes contributing to river salinity are identified, followed by the selection and development of the most appropriate sub-models for each of these based on model purpose, degree of process understanding and data availability, which are then combined to form the hybrid model.

The approach is illustrated for a reach of the River Murray in South Australia. The resulting model consists of five sub-models, including a process-driven in-stream salt transport model, an ANN model to primarily cater to saline groundwater accessions and three linear regression models to account for the flushing of three different waterbodies in the floodplain. Results show that the hybrid model performs very well and is able to capture all variations in salinity with high levels of accuracy. The value of using a hybrid approach is demonstrated by the incremental increase in model performance when different sub-models

are added, as well as the superior performance of the hybrid model compared with that of two benchmark models based on commonly used methods for modelling salinity in rivers, including a process-driven and a data-driven ANN model. In addition to superior predictive performance, the hybrid model results in the development of increased process understanding and is able to be used to assist with the evaluation of various river management options.

Overall, the proposed hybrid approach shows significant promise, although there would be value in applying it to different river systems where different processes dominate and different types of data are available. While the approach has been developed specifically for the modelling of salinity in rivers, there is no reason why some of the underlying principles cannot be applied successfully for other types of hydrological models.

*Data availability:* All sets of original data used in the production of this paper are available publicly from the Surface Water Data System repository at www.waterconnect.sa.gov.au. Readers & reviewers are encouraged to contact the corresponding author directly for any extracted sets of data (e.g. salt load, peak flow duration etc.).

*Competing interests:* The authors declare that they have no conflict of interest.

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
