# Peer review of "Framework for developing hybrid process-driven, artificial neural network and regression models for salinity prediction in river systems"

_Hydrology and Earth System Sciences, 2017_

## Referee Comment (RC1) · Anonymous Referee #1 · 25 Oct 2017

This study developed hybrid process and data-driven model to improve single-driven model performance for modelling salinity in river systems. Despite the paper is well organized and interesting to read, the manuscript in its present form has some weaknesses (mainly lack novelty and scientific findings). General and special comments: (1) The introduction and methodology (13 pages) are too long. Please make it concise and shorter and emphasize the novelty of the study. (2) The results and discussion (1 page excluding tables and figures) are too shorter. Please enrich it and offer more valuable analyses and scientific findings. (3) In Figure 5, the descriptions of "Below 30,000"ãĂĄ"Below 50,000"etc, are imprecise. Please replace it with 'Below 30,000 and above 15,000' etc. Besides, the symbol "sigma" easily causes readers' misunderstanding that the results of Model 1 are equal to the sum of Model 2-5. Please make

major revision for Figure 5. (4) In Eq. (3), so many researchers suggested that it needs use the index Gbench (or Coefficient of Persistence) by replacement of NSE to judge the good-of-fit of the model, when you applied a data-driven model, such as ANN on the basis of benchmark series. Please refer to the reference "Seibert, J. (2001). On the need for benchmarks in hydrological modelling. Hydrological Processes, 15(6), 1063-1064". (5) In Figure 8, the inputs are so important for data-driven model. Why you ANN model just has the exogenous inputs (ex, Lock 5: electrical conductivity with 5-day lag, Lyrup pump station: water level with 3-day lag, Lock 5: flow rate 5-day lag, and Lock 5: water level with 5-day lag), but hasn't the autoregressive input (Lock 4: salinity with 1-day lag). As known, the contributions of autoregressive input for model performance are higher than 80%-90%, however, the contributions of the exogenous inputs for model performance are only 10%-20%. Please explain it. (6) In Figure 8, how do you identify the time-lags of inputs? Please add your methods and results to demonstrate their suitability. (7) Section 2.3, the methodology for identification of most suitable model types is not scientific and imprecise. From the results of Table 2, the most suitable model types are identified based on the degree of data availability and process understanding. How do you quantify the degree of data availability and process understanding? Please make major revision of section 2.3 for enhancing the reliability of this method. (8) In Page 20, lines 438-440: Please add the results of trials with one to four hidden nodes to demonstrate that ANN with three hidden nodes preforms best on the calibration data. Providing the results of RMSE and NSE. (9) In conclusion: you stated the limitation of your methodology "While the approach has been developed specifically for the modelling of salinity in rivers, ….". In fact, this methodology just developed specially for modelling salinity in Murray River of South Australia. Hence, the title of this paper might be changed as follow. Modelling salinity in Murray River of South Australia using hybrid process and data-driven models.

---

## Author Comment (AC1) · 5 Nov 2017

**Response to referee comments on "Modelling salinity in river systems using hybrid process and data-driven models", by Jason M. Hunter et al.**

Note: This document contains the authors' responses to the comments of Referee #1.  The comments made by the referee have been formatted in italic and coloured black, while our responses are upright and coloured blue.

**Response to the comments of Referee #1**

*This study developed hybrid process and data-driven model to improve single-driven model performance for modelling salinity in river systems.  Despite the paper is well organized and interesting to read, the manuscript in its present form has some weaknesses (mainly lack novelty and scientific findings).*

We would like to thank the reviewer for their constructive comments, which will assist with improving the quality of the paper significantly.  Detailed responses to the reviewer's comments are given below.

*(1) The introduction and methodology (13 pages) are too long. Please make it concise and shorter and emphasize the novelty of the study.*

We agree that the novelty of the study could be articulated more clearly.  This will be done in the revised version by:

1. Changing the title of the paper to "Framework for developing hybrid process and data driven (artificial neural network and regression) models of salinity in river systems", thereby highlighting that the primary contribution of the paper is the framework introduced in Section 2, the application of which is illustrated for a real case study in the River Murray, Australia.
2. The fact that the development of the generic framework is the primary contribution of the paper will be highlighted in the revised version of the Abstract.
3. The objectives of the paper, highlighting the contribution of the framework, will be stated explicitly in the Introduction.
4. We will review sections 1 to 3 carefully and make every effort to make them as clear and concise as possible.  For example, the paragraph on lines 72 – 81 will be deleted from the Introduction.

The proposed framework (Section 2) and the demonstration of how this is applied to a real case study (Section 3) are the primary contributions of the paper, so most of the paper is devoted to these topics.  We believe that this is appropriate and will be made clear by the proposed changes to the Title, Abstract and Introduction outlined above.

*(2) The results and discussion (1 page excluding tables and figures) are too shorter.  Please enrich it and offer more valuable analyses and scientific findings.*

As mentioned in our response to Comment (1), the framework and a demonstration of how this is applied to a real case study are the primary contributions of the paper.  In contrast, the purpose of

the actual modelling results is to demonstrate the utility of the proposed approach (or otherwise) and is hence quite brief by design. However, additional discussion in relation to the advantages and disadvantages of the proposed framework, which is the primary contribution of the paper, will be provided in the Results and Discussion section in the revised version of the paper.

45

*(3) In Figure 5, the descriptions of "Below 30,000" A ¸A"Below 50,000"etc, are imprecise. Please replace it with 'Below 30,000 and above 15,000' etc. Besides, the symbol "sigma" easily causes readers' misunderstanding that the results of Model 1 are equal to the sum of Model 2-5. Please make major revision for Figure 5.*

50    The clarity of Figure 5 will be improved in the revised version of the paper in accordance with the reviewer's suggestions.

*(4) In Eq. (3), so many researchers suggested that it needs use the index Gbench (or Coefficient of Persistence) by replacement of NSE to judge the good-of-fit of the model, when you applied a data-*

55    *driven model, such as ANN on the basis of benchmark series. Please refer to the reference "Seibert, J. (2001). On the need for benchmarks in hydrological modelling. Hydrological Processes, 15(6), 1063-1064".*

The goodness-of-fit statistic suggested by the reviewer will be added in the revised version of the paper.

60

*(5) In Figure 8, the inputs are so important for data-driven model. Why you ANN model just has the exogenous inputs (ex, Lock 5: electrical conductivity with 5-day lag, Lyrup pump station: water level with 3-day lag, Lock 5: flow rate 5-day lag, and Lock 5: water level with 5-day lag), but hasn't the autoregressive input (Lock 4: salinity with 1-day lag). As known, the contributions of autoregressive*

65    *input for model performance are higher than 80%-90%, however, the contributions of the exogenous inputs for model performance are only 10%-20%. Please explain it.*

This is a very important point and we would like to thank the reviewer for raising it. Whether autoregressive inputs are considered as candidate inputs or not is a function of the purpose of the model. If the purpose is to obtain the best possible forecasts, then autoregressive inputs should be

70    included as candidate inputs, as suggested by the reviewer. However, if the purpose of the model is to predict an independent variable as a function of other variables, as is the case in the case study considered as the model is supposed to be used to assess the impact of different management options on salinity, then autoregressive inputs cannot be considered.

In relation to the proposed general framework, this point will be added to the discussion on "Model

75    Purpose" in Section 2 (issues like this is the reason for the inclusion of the consideration of model purposes as part of the proposed framework).

In relation to the case study, the reason for not considering autoregressive candidate inputs will be explained in Section 3 in the revised version of the paper.

80    *(6) In Figure 8, how do you identify the time-lags of inputs? Please add your methods and results to demonstrate their suitability.*

We agree that this could be articulated more clearly. In Section 3.6.1, we state that the ANN benchmark model was developed using the same methodology as was used for the development of Model 2, referring to Section 3.5.4 (to ensure the results of the different models can be compared in an objective fashion). In section 3.5.4, we state that the relevant inputs are determined with the aid of correlation analysis. Consequently, the time lags of the inputs in Figure 8 were determined using correlation analysis. However, we will provide model development details in supplementary material for additional clarity and completeness in the revised version of the paper. Given the length of the paper and the primary focus on the proposed hybrid approach, rather than the development of the component models, for which well-developed methodologies already exist, we believe this is more appropriate than giving these details in the paper (which is the reason they were omitted from the first submission).

*(7) Section 2.3, the methodology for identification of most suitable model types is not scientific and imprecise. From the results of Table 2, the most suitable model types are identified based on the degree of data availability and process understanding. How do you quantify the degree of data availability and process understanding? Please make major revision of section 2.3 for enhancing the reliability of this method.*

We agree with the reviewer that ideally, there would be hard and fast rules to assist model developers in determining which model is most appropriate given their circumstances. However, given that we are proposing a generic framework that is designed to be applicable under a wide range of circumstances, this is not possible. The purpose of the proposed framework is to raise these issues as steps that modellers must follow. However, inevitably, a degree of judgement will be required, given the high degree of variability in modelling contexts. In this sense, the concepts introduced are similar to the well-known figure of Grayson and Blöschl (2000) referred to in the paper and shown here, where it is not possible to provide precise quantitative values.

[Figure]

*Figure 1: Relationship Between Data Availability, Model Complexity and Predictive Performance (Grayson and Blöschl, 2000).*

This will be made clearer in the revised version of the paper in a number of places, namely Sections 2.1, 2.3 and 4.

Reference:

Grayson, R. B., and Blöschl, G.: Spatial Patterns in Catchment Hydrology: Observations and Modelling, Cambridge University Press, Cambridge, United Kingdom, 2000.

115

*(8) In Page 20, lines 438-440: Please add the results of trials with one to four hidden nodes to demonstrate that ANN with three hidden nodes preforms best on the calibration data. Providing the results of RMSE and NSE.*

These results will be provided as supplementary material in the revised version of the paper. As
120    mentioned in our response to Comment (6), we believe this to be most appropriate, given the length and focus of the paper.

*(9) You stated the limitation of your methodology "While the approach has been developed specifically for the modelling of salinity in rivers, ….".  In fact, this methodology just developed*
125    *specially for modelling salinity in Murray River of South Australia. Hence, the title of this paper might be changed as follow.  Modelling salinity in Murray River of South Australia using hybrid process and data-driven models.*

We agree that the purpose and contribution of the paper was not articulated as clearly as it should
have been.  However, as per our responses to Comments (1) and (2), the paper introduces a generic
130    framework that is illustrated using the River Murray case study.  This will be made clear by the
changed title and clearly stated objectives (see responses to Comment (1)).

---

## Referee Comment (RC2) · Anonymous Referee #2 · 22 Feb 2018

The paper presents a well thought out and executed approach to model salinity in a complex river environment.

---

## Author Response (AR1)

**Response to referee comments on "Modelling salinity in river systems using hybrid process and data-driven models", by Jason M. Hunter et al.**

Note: This document contains the authors' responses to the comments of Referee #1.  The comments made by the referee have been formatted in italic, bold and coloured black, while our responses are upright and coloured blue.  Quotations from the paper in question are coloured blue, italicised, and indented.

**Response to the comments of Referee #1**

***This study developed hybrid process and data-driven model to improve single-driven model performance for modelling salinity in river systems.  Despite the paper is well organized and interesting to read, the manuscript in its present form has some weaknesses (mainly lack novelty and scientific findings).***

We would like to thank the reviewer for their constructive comments, which have assisted with improving the quality of the paper significantly.  Detailed responses to the reviewer's comments are given below.

***(1) The introduction and methodology (13 pages) are too long. Please make it concise and shorter and emphasize the novelty of the study.***

We agree that the novelty of the study could be articulated more clearly.  This has been done in the revised version by:

1. Changing the title of the paper to "Framework for developing hybrid process-driven, artificial neural network and regression models for salinity prediction in river systems", thereby highlighting that the primary contribution of the paper is the framework introduced in Section 2, the application of which is illustrated for a real case study in the River Murray, Australia.

2. The fact that the development of the generic framework is the primary contribution of the paper has been highlighted in lines 15-16, in the revised version of the Abstract.

   *"In order to overcome these limitations, a generic framework for developing hybrid process and data-driven models of salinity in river systems is introduced and applied in this paper."*

3. The objectives of the paper, highlighting the contribution of the framework, have been stated explicitly in the Introduction, in lines 98-136.

   *"In addition, these studies have focused on a particular hybrid model structure, rather than a generic framework that can be used to develop the most suitable hybrid model in different settings.  In order to overcome the above shortcomings, the objectives of this paper are:*

   *1. To introduce a generic, high level conceptual framework for the development of hybrid models for modelling salinity in river systems that uses a combination of model purpose, knowledge of underlying system processes, and type and amount of available data to provide guidance for the selection of the suitable sub-models, thereby enabling the most appropriate balance between*

*hypothetic and data influence to be struck in their development. While the proposed approach is specific to salinity modelling, the underlying principles presented are likely to be more widely applicable. The modelling of salinity in river systems is selected as the focus of the approach, as:*

a. *High levels of salinity are a major concern in many river systems around the world (Rengasamy, 2006), due to their potentially detrimental impacts on the growth of agricultural crops, vegetation, bacteria and algae (Hart et al., 1991; Maier and Dandy, 1996) and adverse effects on water quality, as well as the stability of freshwater and neighbouring ecosystems. In addition, high salinity levels can have significant negative financial consequences stemming from the ongoing expense of treating drinking water, pumping at groundwater interception wells and from diminishing agricultural returns (Moxey, 2012).*

b. *Salinity in river systems is generally affected by a number of complex processes (Williams, 2001; Goss, 2003), and the degree to which these processes are understood varies significantly (Maier and Dandy 1996; Woods, 2015). For example, there is generally a good understanding of the processes involved in the transport of salt with discharge, as salt is a conservative constituent. However, understanding of the complex processes associated with the accession of additional salt loads into the main river channel is often limited, as they are generally influenced by multiple interacting factors (e.g. land use, historical inundation regime, surface water-groundwater interactions, abstraction / recharge processes). In addition, the data needed to support the development of different types of models is highly variable.*

c. *Current efforts directed towards the modelling of salinity in river systems has generally relied on either process-driven (Banerjee et al., 2011; Habib et al., 2007; Liu et al., 2007; Woods, 2015) or data-driven (Maier and Dandy, 1996; Huang and Foo, 2002; Suen and Lai, 2013; Bowden et al., 2005a; Bowden et al., 2002; Kingston et al., 2005; Rath et al., 2017) approaches. This has resulted in a number of limitations, such as difficulties in modelling the accession of salt via groundwater, wetlands and floodplains (e.g. groundwater regime shifts and flushing) explicitly (Harrington et al., 2006), which in turn makes it difficult to understand the relative importance of the different sources of saline accessions and to assess the potential utility of some of the management options mentioned earlier.*

2. *To illustrate the application and test the utility of the framework by applying it to a reach of the River Murray in South Australia, as this is an area where improved salinity modelling for management purposes would be of significant benefit (Beecham et al., 2003)."*

4. Sections 1 to 3 have been reviewed carefully and every effort has been made to make them as clear and concise as possible. For example, the paragraph at line 73 in the revised version has been deleted from the Introduction (previously lines 76-85, as below).

*"It should be noted that in recent times, the use of flexible, hypothetically influenced model frameworks has been suggested as a means of matching appropriate model processes / complexity with available data (e.g. Clark et al., 2011; Fenicia et al., 2011; Kavetski and Fenicia, 2011). In these types of models, understanding of underlying system processes is used to suggest model structure components that provide conceptual representations of potentially relevant processes, whereas the available data are used to determine which of these are most appropriate. It should also be noted that although the structure of data-driven models is dictated purely by the available data, care needs to be taken that appropriate methods are used to reduce model complexity as much as possible in order to avoid overfitting (e.g. Kingston et al., 2008; Galelli et al., 2014; Wu et al., 2014). In other words, even for data-driven models, the available data should dictate the degree of model complexity that can be supported. This is particularly the case for models where the number of model parameters requiring calibration can be large, such as artificial neural networks (Elshorbagy et al., 2010; Maier et al., 2010; Abrahart et al., 2012)."*

The proposed framework (Section 2) and the demonstration of how this is applied to a real case study (Section 3) are the primary contributions of the paper, so most of the paper is devoted to these topics. We believe that this is appropriate and has now been made clear by the proposed changes to the Title, Abstract and Introduction outlined above.

**(2) The results and discussion (1 page excluding tables and figures) are too shorter. Please enrich it and offer more valuable analyses and scientific findings.**

As mentioned in our response to Comment (1), the framework and a demonstration of how this is applied to a real case study are the primary contributions of the paper. In contrast, the purpose of the actual modelling results is to demonstrate the utility of the proposed approach (or otherwise) and is hence quite brief by design. However, additional discussion in relation to the advantages and disadvantages of the proposed framework, which is the primary contribution of the paper, have been provided in the Results and Discussion section in the revised version of the paper, in lines 650 - 654.

*"Overall, the results of the illustrative case study highlight the potential benefits of the proposed framework. By considering the relevant processes affecting river salinity at the site of interest, how well they are understood and can be represented mathematically, how much data there is to support model development and what the primary purpose of the model is, a hybrid model was able to be developed that not only results in better predictive performance than the corresponding benchmark process- and data-driven models, but is also more useful from a management perspective."*

*(3) In Figure 5, the descriptions of "Below 30,000" A ̧A"Below 50,000"etc, are imprecise.  Please replace it with 'Below 30,000 and above 15,000' etc. Besides, the symbol "sigma" easily causes readers' misunderstanding that the results of Model 1 are equal to the sum of Model 2-5.  Please make major revision for Figure 5.*

135     The clarity of Figure 5 has been improved in the revised version of the paper in accordance with the reviewer's suggestions.

[Figure]

*Figure 1: Figure 5 from the original submission, to be amended.*

[Figure]

    *Figure 2: Figure 5 in the revised submission.*

**(4) In Eq. (3), so many researchers suggested that it needs use the index Gbench (or Coefficient of Persistence) by replacement of NSE to judge the good-of-fit of the model, when you applied a data-driven model, such as ANN on the basis of benchmark series. Please refer to the reference**
145    **"Seibert, J. (2001). On the need for benchmarks in hydrological modelling. Hydrological Processes, 15(6), 1063-1064".**

The goodness-of-fit statistic suggested by the reviewer has been added in the revised version of the paper, in lines 390-304 and lines 633-639.

150    *"The goodness of fit of the hybrid model is evaluated against two benchmark models, one process-driven and one data-driven, with the metric $G_{bench}$ introduced in Seibert, 2001 (Eq. 4).*

$$G_{bench} = 1 - \frac{\sum_{i=1}^{n}\left(x_o^i - x_m^i\right)^2}{\sum_{i=1}^{n}\left(x_o^i - x_b^i\right)^2} \qquad (4)$$

- *$x_b$ = benchmark model data points."*

155 *"To reinforce and clarify the differences between the hybrid model and the benchmark model, the $G_{bench}$ index was calculated (Eq. 4).  The $G_{bench}$ index is structured similar to the NSE, but replaces the mean of the observed time series with the time series of a benchmark model, so an index of zero indicates that model performance is equal to that of the benchmark model.  The values of the $G_{bench}$ index are 0.36 and 0.90 for the data-driven and process-driven benchmark models respectively.  Both values are*
160 *positive, indicating that the performance of the hybrid model is better than the performance of either benchmark model, with it performing better against the process-driven benchmark model than the data-driven benchmark model."*

**(5) In Figure 8, the inputs are so important for data-driven model.  Why you ANN model just has**
165 **the exogenous inputs (ex, Lock 5: electrical conductivity with 5-day lag, Lyrup pump station: water level with 3-day lag, Lock 5: flow rate 5-day lag, and Lock 5:  water level with 5-day lag), but hasn't the autoregressive input (Lock 4: salinity with 1-day lag).  As known, the contributions of autoregressive input for model performance are higher than 80%-90%, however, the contributions of the exogenous inputs for model performance are only 10%-20%.  Please explain it.**

170 This is a very important point and we would like to thank the reviewer for raising it.  Whether autoregressive inputs are considered as candidate inputs or not is a function of the purpose of the model.  If the purpose is to obtain the best possible forecasts, then autoregressive inputs should be included as candidate inputs, as suggested by the reviewer.  However, if the purpose of the model is to predict an independent variable as a function of other variables, as is the case in the case study
175 considered, as the model is supposed to be used to assess the impact of different management options on salinity, then autoregressive inputs cannot be considered.

In relation to the proposed general framework, this point has been added to the discussion on "Model Purpose" in Section 2, in lines 196-200 (issues like this are the reason for the inclusion of the consideration of model purposes as part of the proposed framework).

180 *"…This also has an impact on which potential model inputs are considered.  For example, if forecasting is the primary model purpose, auto-regressive values of the model output should be considered as potential inputs (e.g. Bowden et al., 2005b), as this is likely to improve the quality of the forecasts.  In contrast, if the purpose of the model is to assess the impact of different management options on salinity,*
185 *autoregressive values of the model output cannot be considered as potential model inputs, as the model output has to be independent of the model input(s) in such cases."*

In relation to the case study, the reason for not considering autoregressive candidate inputs has
190 been explained in Section 3 (lines 426-429) in the revised version of the paper.

*"It should be noted that past values of the model output are not considered as potential model inputs, as the purpose of the model is to assess the impact of different management options on river salinity, rather than forecasting salinity future salinity values (see Section 2.3)."*

195

**(6) In Figure 8, how do you identify the time-lags of inputs?  Please add your methods and results to demonstrate their suitability.**

We agree that this could be articulated more clearly.  In Section 3.6.1, we state that the ANN benchmark model was developed using the same methodology as was used for the development of Model 2, referring to Section 3.5.4 (to ensure the results of the different models can be compared in an objective fashion).  In section 3.5.4, we state that the relevant inputs are determined with the aid of correlation analysis.  Consequently, the time lags of the inputs in Figure 8 were determined using correlation analysis.  However, we have provided model development details in supplementary material for additional clarity and completeness in the revised version of the paper.  Given the length of the paper and the primary focus on the proposed hybrid approach, rather than the development of the component models, for which well-developed methodologies already exist, we believe this is more appropriate than giving these details in the paper (which is the reason they were omitted from the first submission).

**(7)  Section 2.3, the methodology for identification of most suitable model types is not scientific and imprecise.  From the results of Table 2, the most suitable model types are identified based on the degree of data availability and process understanding.  How do you quantify the degree of data availability and process understanding? Please make major revision of section 2.3 for enhancing the reliability of this method.**

We agree with the reviewer that ideally, there would be hard and fast rules to assist model developers in determining which model is most appropriate given their circumstances.  However, given that we are proposing a generic framework that is designed to be applicable under a wide range of circumstances, this is not possible.  The purpose of the proposed framework is to raise these issues as steps that modellers must follow.  However, inevitably, a degree of judgement will be required, given the high degree of variability in modelling contexts.  In this sense, the concepts introduced are similar to the well-known figure of Grayson and Blöschl (2000) referred to in the paper and shown here, where it is not possible to provide precise quantitative values.

[Figure]

*Figure 3: Relationship Between Data Availability, Model Complexity and Predictive Performance (Grayson and Blöschl, 2000).*

This has been made clearer in the revised version of the paper in a number of places, namely Sections 2.1 (lines 155-156), 2.3 (lines 216-219)  and 4 (lines 654-656), as shown below:

*"Given the conceptual nature of the framework, it provides high-level guidance and there is some subjectivity in its application to a particular case study."*

 *"It is important to note that the proposed framework is conceptual in nature and designed to provide high-level guidance.  Consequently, its implementation for particular case studies is subjective.  For example, how much data is required to support a particular modelling approach is case study dependent and relies on the judgement of the model developer.  Consequently, this stage of the process may be iterative."*

*"However, given the conceptual nature of the proposed framework and the level of subjectivity required to implement it, it is not possible to tell if an even better model could have been developed had different decisions been made with regard to model types."*

These results have been provided as supplementary material in the revised version of the paper.  As mentioned in our response to Comment (6), we believe this to be most appropriate, given the length and focus of the paper.

255  *(9) You stated the limitation of your methodology "While the approach has been developed specifically for the modelling of salinity in rivers, ....".   In fact, this methodology just developed specially for modelling salinity in Murray River of South Australia.  Hence, the title of this paper might be changed as follow.  Modelling salinity in Murray River of South Australia using hybrid process and data-driven models.*

260  We agree that the purpose and contribution of the paper was not articulated as clearly as it should have been.  However, as per our responses to Comments (1) and (2), the paper introduces a generic framework that is illustrated using the River Murray case study.  This has been made clear by the changed title and clearly stated objectives (see responses to Comment (1)).

265

**Response to the comments of Referee #2**

  *(1) The paper presents a well thought out and executed approach to model salinity in a complex*
270  *river environment.*

  We are very grateful to the reviewer for their kind words, and for their recommendation that the paper be accepted as is.

[revised manuscript text omitted]

75    ~~It should be noted that in recent times, the use of flexible, hypothetically influenced model frameworks has been suggested as a means of matching appropriate model processes / complexity with available data (e.g. Clark et al., 2011; Fenicia et al., 2011; Kavetski and Fenicia, 2011). In these types of models, understanding of underlying system processes is used to suggest model structure components that provide conceptual representations of potentially relevant processes, whereas the available data are used to determine which of these are most appropriate. It should also be noted that although the structure of data-driven models~~

80    ~~is dictated purely by the available data, care needs to be taken that appropriate methods are used to reduce model complexity as much as possible in order to avoid overfitting (e.g. Kingston et al., 2008; Galelli et al., 2014; Wu et al., 2014). In other words, even for data-driven models, the available data should dictate the degree of model complexity that can be supported. This is particularly the case for models where the number of model parameters requiring calibration can be large, such as artificial neural networks (Elshorbagy et al., 2010; Maier et al., 2010; Abrahart et al., 2012).~~

[revised manuscript text omitted]

neighbouring ecosystems. In addition, high salinity levels can have significant negative financial consequences stemming from the ongoing expense of treating drinking water, pumping at groundwater interception wells and from diminishing agricultural returns (Moxey, 2012). ~~However, development of appropriate models can assist with addressing these issues via a range of management techniques, including the control of changing land uses (Foley et al., 2005), optimising water extraction timing (Maier and Dandy, 1996), engineered flows and dilution management (Young, 2000) and determination of the optimal location and timing of salt interception schemes (Tefler et al., 2012).~~

[revised manuscript text omitted]